# STREAM RAG: INSTANT AND ACCURATE SPOKEN DIALOGUE SYSTEMS WITH STREAMING TOOL USAGE

## ABSTRACT

End-to-end speech-in speech-out dialogue systems are emerging as a powerful alternative to traditional ASR–LLM–TTS pipelines, generating more natural, expressive responses with significantly lower latency. However, these systems remain prone to hallucinations due to limited factual grounding. While text-based dialogue models have effectively mitigated this issue through tools such as web search and knowledge-graph APIs, extending such capabilities to speech-in speech-out systems remains underexplored. A key challenge is that tool integration substantially increases response latency, disrupting conversational flow. To mitigate this, we propose Streaming Retrieval-Augmented Generation (*Stream RAG*), a novel framework that reduces user-perceived latency by predicting tool queries in parallel with user speech, even before the user finishes speaking. Specifically, we develop a post-training pipeline that teaches the model when to *issue tool calls* during ongoing speech and how to generate spoken summaries that fuse audio queries with retrieved text results, thereby improving both *accuracy and responsiveness*. To evaluate our approach, we construct AudioCRAG, a benchmark created by converting queries from the publicly available CRAG dataset into speech form. Experimental results demonstrate that our *Stream RAG* approach increases QA accuracy by over 200% relative and further enhances user experience by reducing tool use latency by 17%. Importantly, our *Stream RAG* approach is modality-agnostic and can be applied equally to typed input, paving the way for more agentic, real-time AI assistants.

## 1 INTRODUCTION

Spoken Dialogue Systems (SDS) are foundational to many everyday technologies, powering intelligent assistants such as Alexa and Siri, as well as interactive voice response systems in customer service. With the rapid expansion of SDS capabilities to mobile phones and wearable devices, the need for robust, scalable, and generalizable solutions has never been greater. Traditionally, SDS have relied on cascaded pipelines composed of multiple modules—including voice activity detection (VAD), automatic speech recognition (ASR), natural language understanding (NLU), natural language generation (NLG), and text-to-speech (TTS) synthesis—each introducing potential points of failure and latency (Glass, 1999; Huang et al., 2024). Recently, end-to-end (E2E) SDS (Xie & Wu, 2024; Nguyen et al., 2023; Meng et al., 2024; Zhang et al., 2024; Arora et al., 2025b) have been proposed, which directly generate spoken responses from speech input within a unified architecture. This E2E approach not only mitigates error propagation across modules but also captures non-phonemic information more effectively, resulting in significantly lower inference time and computational overhead, and paving the way for more natural and efficient conversational experiences.

Despite these advances, current E2E SDS are fundamentally constrained by their reliance on internalized knowledge from static training data, which often results in responses that lack factual grounding or fail to reflect the most up to date information. This shortcoming is particularly critical for action-oriented or knowledge-seeking tasks, such as booking hotels or answering questions about current events. In contrast, text-based conversational assistants have begun to overcome these limitations by integrating external tools through Retrieval-Augmented Generation (RAG)(Yang et al., 2024; Chen et al., 2023a;c; Gao et al., 2024a;b), dynamically retrieving relevant information from sources like web search, knowledge graphs (KG), and real-time APIs. Yet, the integration of such tool use into E2E SDS remains largely unexplored. A key challenge is that while external tools can

substantially improve factual accuracy, invoking them often introduces additional latency, leading to awkward silences that disrupt the natural conversational flow. *How can we trade-off between accuracy and responsiveness for developing SDS that feel both intelligent and natural?*

In this paper, we present, to the best of our knowledge, the first *speech-in, speech-out language model that seamlessly integrates external tool invocation with low latency*. The key idea is a **Stream RAG** strategy that generates tool queries in parallel with user speech, often times even before the user has finished speaking (Fig. 1). A naive implementation of streaming queries, however, faces two challenges: (1) queries issued from partial speech may be suboptimal, yielding distracting tool outputs and inaccurate responses; and (2) unnecessary tool calls may be triggered, wasting computational resources. We introduce effective modeling techniques to address these challenges and make the following contributions.

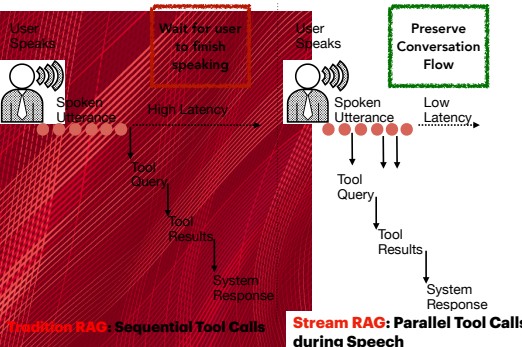

Figure 1: Comparison of Traditional RAG with proposed Stream RAG which fires tool queries in parallel with user speech.

**Contribution 1**: We introduce a formal framework for tool integration in speech-in speech-out systems and empirically show that leveraging web search and KG APIs significantly enhances factual question answering. Evaluating three state-of-the-art (SOTA) models, Qwen-OMNI (Xu et al., 2025), OpusLM (Tian et al., 2025), and Kimi-Audio (Ding et al., 2025), we find that external tool integration delivers substantial performance gains, boosting accuracy by up to 140% relative. However, tool usage also introduces considerable latency, increasing first-token response time by 2.3x.

**Contribution 2**: To address this, we propose *Streaming Retrieval-Augmented Generation* (*Stream RAG*), the *first* framework that empowers the system to trigger tool queries in parallel with user speech, even before the user finishes speaking. Within this framework, we introduce two novel approaches: (1) *Fixed-Interval Stream RAG*, which issues tool queries at regular intervals during speech input and carefully examines quality of retrieval results on the full query to guarantee response quality, and can be incorporated into any speech-in, speech-out model without post-training; (2) *Model-Triggered Stream RAG*, which post-trains the model to intelligently determine optimal query timing based on the evolving user utterance to save computation resources. Our results demonstrate that our proposed *Model-Triggered Stream RAG* delivers over 200% relative improvement in accuracy (from 11.1% to 34.2% absolute in T. 1) compared to the no-tool baseline, while also reducing tool result generation latency by 17%. Though designed for speech-in speech-out systems, stream RAG can also be adapted in cascaded SDS, or even chatbots as users type.

**Contribution 3**: Finally, we introduce AudioCRAG, a benchmark created by recording spoken queries from the CRAG (Yang et al., 2024) dataset, enabling robust evaluation of tool usage capabilities in speech-in speech-out systems. To support future research, we will open source our training code, supporting future research in tool-integrated voice assistants.

## 2 RELATED STUDIES

### 2.1 BENCHMARKS FOR TOOL USAGE

Recent advances in benchmarking text-based dialogue systems for tool usage (Chen et al., 2023b; Ouyang et al., 2025; Cheng & Dou, 2025; Cohen et al., 2025; Xiong et al., 2024a) have primarily focused on evaluating factual question answering and task completion within simulated environments (detailed related work discussion in Appendix A.1, A.2). The CRAG benchmark (Yang et al., 2024) is a leading example, featuring 4,409 question-answer pairs and providing mock APIs for both web and KG search. Recent benchmarks (Meta CRAG-MM Challenge Organizers, 2025; Ma et al., 2024; Jang et al., 2025) have extended tool-augmented dialogue evaluation to multimodal input and longer-context scenarios. While these benchmarks have significantly advanced the evaluation of tool-augmented dialogue systems, they remain largely limited to text-based outputs and do not fully address the unique challenges presented by speech-in speech-out systems.

## 2.2 E2E SPOKEN DIALOGUE SYSTEMS

Several E2E spoken dialogue systems (Xu et al., 2025; Xie & Wu, 2024; Arora et al., 2025a; Nguyen et al., 2023; Meng et al., 2024; Zhang et al., 2024; Arora et al., 2025b) have recently been introduced, demonstrating impressive semantic understanding and high audio quality in their responses. However, these systems have not yet been trained or evaluated for their ability to *use external tools*. Another research direction (Feng et al., 2025) explores E2E RAG for direct speech-to-text retrieval, utilizing multimodal embeddings for enabling speech utterances to directly retrieve relevant text. Although recent studies (Maben et al., 2025) have developed web interfaces that integrate tools into speech-in speech-out scenarios using cascaded pipelines, comprehensive empirical investigations of E2E speech-in speech-out systems and, crucially, systematic analyses of user-perceived latency, remain largely unexplored. In this work, we address these gaps by developing a comprehensive framework for tool integration in E2E speech-in speech-out systems and designing benchmarks to quantitatively assess the tool usage capabilities of SOTA models. Another work, WavRAG (Chen et al., 2025), extends RAG to speech-in speech-out models using audio–text fusion. However, like prior RAG approaches, it performs retrieval only after endpoint detection. This retrieve-after-endpoint paradigm introduces substantial latency when tool calls require queries over 100K web documents. Hence we study the problem of streaming tool use and introduce Stream RAG, the *first* framework that enables models to issue partial tool queries during ongoing speech to mask external API latency.

## 3 METHODOLOGY

A RAG spoken conversation system takes an audio question $Q$ as input and outputs a spoken answer $A$. Let the ASR transcript of audio question be $X^{\mathrm{asr}}$ and the ASR transcript of the spoken answer be $X^{\mathrm{res}}$. Answers are generated by speech-in speech-out models, leveraging both the model's internal knowledge and information retrieved from external sources. To incorporate external information, the model needs to formulate a tool query $Q^T$ to retrieve relevant results $R$ from an external tool $T$.

### 3.1 TOOL INTEGRATION FOR SPEECH-IN SPEECH-OUT LLMS

Figure 2 illustrates our proposed formulation for integrating external tools into speech-in speech-out systems. We introduce a two-stage inference approach: *Query Generation* and *Response Generation*. In the Query Generation stage, the system processes an audio question and generates queries for each external tool to retrieve relevant information by maximizing the posterior distribution $P(Q^T|Q)$ (Examples of generated queries are provided in T. 7 in the Appendix.). In the Response Generation stage, the retrieved results $R$ from these tools are combined with the original audio question and input into the model to generate the final spoken response by maximizing the posterior distribution $P(A|Q,R)$. By conditioning the final output generation on the input audio, this formulation not only provides a simple and effective

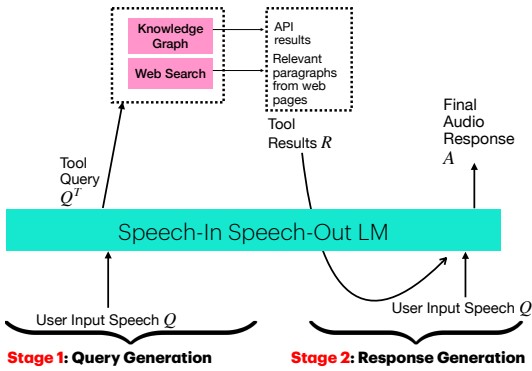

Figure 2: Proposed formulation for integrating tool usage in E2E speech-in, speech-out dialogue systems using a two-stage inference approach.

mechanism for interacting with text-based APIs, but also preserves the key advantages of speech-in speech-out systems, mitigating error propagation and enabling the model to capture non-phonemic information (such as prosody and speaker intent) more effectively.

### 3.2 STREAM RAG

RAG-based systems, as proposed in S. 3.1, can significantly improve factual accuracy by incorporating external tools. However, these tool calls often introduce substantial latency, which is particularly problematic in speech-in, speech-out applications where users expect rapid, conversational responses and even brief silences can disrupt the natural flow of dialogue. One way to address this

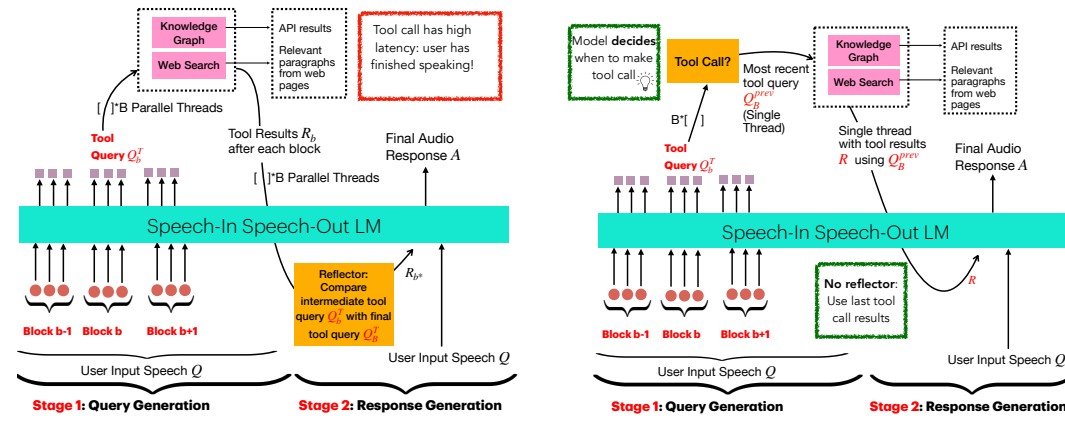

Figure 3: Proposed formulations for streaming tool query generation, referred to as *Stream RAG*, to minimize user-perceived latency in speech-in, speech-out systems. (a) Fixed-Interval Stream RAG: tool calls are triggered at fixed intervals and evaluated by a reflector module. (b) Model-Triggered Stream RAG: the model autonomously decides when to make tool calls, eliminating the need for a reflector and directly utilizing the most recent tool results for response generation.

challenge lies in the nature of audio inputs, which arrive as a continuous stream. This streaming property enables tool calls to be initiated before the user has finished speaking, offering a unique opportunity to mitigate latency.

To minimize user-perceived latency, we introduce *Stream RAG*: the first framework to generate and issue tool queries in parallel as audio input is received. This novel approach is built on three key design components: ① Trigger: When to initiate a new tool query; ② Threads: The number of parallel tool query threads; ③ Reflector: The module that determines whether intermediate tool results are sufficient for generating the final output. By exploring different design choices for each component, we introduce two complementary approaches for streaming tool query generation in the following subsections: a fixed-interval trigger method and a model-based trigger method.

### 3.2.1 FIXED-INTERVAL STREAM RAG

In this approach, the *trigger* is set to fire tool calls at fixed chunk intervals during audio input. The input speech $Q$ is divided into a sequence of $B$ blocks, $Q = \{Q_b \mid b = 1, \ldots, B\}$, with each block containing $N_{\text{block}}$ frames. To approximate $P(Q^T|Q)$ as described in S. 3.1, we follow a block-wise prediction strategy. In this approach, after processing each audio block $b$, the model predicts a tool query $\hat{Q}_b^T$ by conditioning on the input speech accumulated up to block b, specifically $Q_{1:b}$:

$$\hat{Q}_b^T = \arg\max_{Q_b^T} P(Q_b^T|Q_{1:b}) \tag{1}$$

This strategy results in $B$ parallel tool call threads running simultaneously (see Figure 3a), where each thread generates a tool query prediction $\hat{Q}_b^T$ for its corresponding block $b \in [1, B]$. The tool queries $\hat{Q}_b^T$ generated after each block are then stored in cache. Given the high latency of tool calls (See T. 3), users typically complete their utterances before tool responses are ready. Upon utterance completion, an explicit *reflector* module "reflect()" evaluates the cached intermediate queries $\hat{Q}_b^T$ against final query $\hat{Q}_B^T$ to determine whether an early intermediate tool call provides sufficient information to answer the user's question $Q$. The reflector module systematically evaluates all intermediate queries in the cache and identifies the *earliest* sufficient tool call $b^\star$ where the intermediate query $\hat{Q}_b^T$ will give the same result as final query $\hat{Q}_B^T$ as shown:

$$b^\star = \min\{b \in [1, B] \text{ where reflect}(\hat{Q}_b^T, \hat{Q}_B^T) = \text{True}\}. \tag{2}$$

All subsequent parallel tool calls after $b^\star$ are promptly terminated, and the retrieved results $R_{b^\star}$ from this intermediate call $\hat{Q}_{b^\star}^T$ are used to generate the final spoken response $A$ by maximizing the posterior distribution $P(A|Q, R_{b^\star})$ (instead of $P(A|Q, R)$ in S. 3.1). We employ a reflector module that uses simple yet effective heuristics: (a) if the top 5 web documents for an intermediate

web query match those of the final web query, and (b) if the KG results for intermediate and final KG queries are identical. These heuristics ensure that the information retrieved from an early tool call using $\hat{Q}_{b*}^T$ is consistent with what would have been obtained by waiting for the final tool call using $\hat{Q}_B^T$, thereby providing a strong quality guarantee. Since most tool call latency arises from tool results generation i.e. chunking and reranking the chunks of web documents, these checks enable significant latency savings without any compromise in model performance. A key advantage of this strategy is its plug-and-play nature: it requires no additional post-training for speech-in, speech-out models and can be directly applied at inference time across a variety of architectures. However, there are important considerations. First, generating parallel tool calls at every fixed interval (Eq. 1) increases computational overhead, which may pose challenges for deployment on resource-constrained devices such as wearables devices. Second, reliance on an external reflector module (Eq. 2), which must still process the final input block to confirm the last query and judge whether intermediate results are sufficient, may limit the extent of achievable latency improvements.

### 3.2.2 MODEL-TRIGGERED STREAM RAG

To address the limitations of Fixed-Interval Stream RAG and further optimize both efficiency and responsiveness, we propose a more adaptive approach: *Model-Triggered Stream RAG*. Here, the *trigger* is learned: the model is trained to autonomously determine the optimal moments to initiate tool queries, issuing a query only when it encounters new or additional information as illustrated in Figure 3b. In this formulation, the model receives user input in fixed chunk intervals as before and intelligently determines whether a tool call is needed after each block $b$. The model can either: ① Predict NO_QUERY if a new tool query is unnecessary, or ② Generate a new tool query. To make this decision, the model conditions on both the accumulated input speech $Q_{1:b}$ (see Eq. 1) as well as most recent tool query $\hat{Q}_b^{\text{prev}} = \hat{Q}_{\max\{\,i<b:\hat{Q}_i^T \neq \text{NO\_QUERY}\,\}}^T$ as shown:

$$\hat{Q}_b^T = \arg\max_{Q_b^T} P(Q_b^T | Q_{1:b}, \hat{Q}_b^{\text{prev}}) \tag{3}$$

When a new query $\hat{Q}_b^T \neq \text{NO\_QUERY}$, the system immediately terminates any ongoing tool calls for the previous query $\hat{Q}_b^{\text{prev}}$, ensuring that only a *single tool call thread* runs at any given time. (Examples of generated $\hat{Q}_b^T$ are provided in T. 8 in the Appendix.) This approach offers several key advantages. First, it effectively eliminating redundant parallel threads and significantly reducing computational overhead. This is especially important for deployment on resource-constrained devices. Second, this formulation removes the need for an external *reflector* (Eq. 2) module. The model confidently relies on the results $R$ from the most recent tool call using $\hat{Q}_B^{\text{prev}}$ to generate the spoken response $A$ by maximizing $P(A|Q, R)$, reducing system complexity.

**Post-training**: To train the model, we transform text-based tool usage benchmarks into spoken format (See S. A.7). Word-level timestamps are computed using a pre-trained ASR model. For each partial ASR transcript $X_b^{\text{asr}}$ up to block $b$, we generate corresponding queries for each tool $\overline{Q_b^T}$ using an LLM as pseudo ground truth (GT). To create effective training labels that teach the model when to trigger new queries, we employ a similarity-based labeling strategy where we compare the current pseudo GT query $\overline{Q_b^T}$ with the most recent non-empty tool query label $\hat{Q}_b^{\text{prev}}$ (see Eq. 3) before block $b$. Our labeling function assigns the training label $\hat{Q}_b^T$ (Eq. 3) for the tool query after block $b$ as follows: ① when the current query is sufficiently similar to the previous query (as determined by manually defined heuristics $f(\cdots)$), we assign the special label NO_QUERY to teach the model that no new tool call is needed. ② When the queries are sufficiently different, we assign the actual pseudo ground truth query $\overline{Q_b^T}$ as the label to teach the model to trigger a new tool call:

$$\hat{Q}_b^T = \begin{cases} \text{NO\_QUERY}, & \text{if } f(\overline{Q_b^T}, \hat{Q}_b^{\text{prev}}) = \text{True}, \\ \overline{Q_b^T}, & \text{else .} \end{cases} \tag{4}$$

For KG queries, we assign a NO_QUERY label when the current query exactly matches the previous one. For web queries, we assign a NO_QUERY label if the top five retrieved documents for the current query remain unchanged from the previous query.

We employ a multi-task fine-tuning strategy targeting two key capabilities. First, we train the model on Streaming Tool Query Generation by optimizing $P(Q_b^T | Q_{1:b}, \hat{Q}_b^{\text{prev}})$ for $b \in [1, B]$, enabling in-

Table 1: Performance comparison of accuracy, first-token and tool use latency for three models—Qwen2.5-7B, OpusLM, and Kimi Audio—across three settings: Closed Book (without tool usage), Open Book (with tool usage), and Stream RAG (*Model-Triggered Stream RAG*). We evaluate all models on both the AudioCRAG-Synthetic (Syn.) and AudioCRAG-Human (Hum.). Stream RAG is not applied to Kimi-Audio as it can handle only a restricted length of tool result references (S. A.6). *: OpusLM currently does not support taking tool result references in speech-out settings in zero-shot.

| Setting | Ref length | Model | Accuracy | | Latency | | | |
| | | | | | First Token (s) | | Tool Use (s) | |
| | | | Syn. | Hum. | Syn. | Hum. | Syn. | Hum. |
|---|---|---|---|---|---|---|---|---|
| Closed Book | 0 | Qwen2.5-7B | 11.1 | 13.1 | 1.34 | 1.24 | ✗ | ✗ |
| | 0 | OpusLM | 18.4 | 15.5 | 5.67 | 7.07 | ✗ | ✗ |
| | 0 | Kimi Audio | 16.7 | 16.0 | 0.85 | 0.89 | ✗ | ✗ |
| Open Book | 23K | Qwen2.5-7B | 26.3 | 26.9 | 5.90 | 5.40 | 3.37 | 3.37 |
| (S. 3.1) | 15K | OpusLM* | 0.0 | 0.0 | 9.05 | 10.44 | 3.97 | 3.97 |
| | 500 | Kimi Audio | 21.8 | 19.6 | 4.22 | 4.22 | 3.15 | 3.15 |
| Stream RAG | 23K | Qwen2.5-7B | 34.2 | 37.4 | 5.32 | 3.60 | 2.79 | 1.57 |
| (S. 3.2.2) | 15K | OpusLM | 23.6 | 22.8 | 8.63 | 9.04 | 3.55 | 2.57 |

telligent decisions about when to trigger tool queries. Second, we fine-tune on Response Generation by optimizing $P(A|Q, R)$ (S. 3.1) to improve the intelligibility of the speech output.

An important aspect of our post-training is enhancing the model's ability to recover from errors in intermediate query predictions. For example, when presented with the audio question, "Who founded Rare Beauty in 2019?", we observed that an initial misinterpretation of $\hat{Q}_b^{\text{prev}}$, such as "Red Bull founder", can lead the model to subsequently generate NO_QUERY labels, effectively halting further attempts to retrieve the correct information. This issue arises because, during training, the model is always provided with correct previous query $\hat{Q}_b^{\text{prev}}$, whereas during inference, it may make errors due to partial utterances $Q_{1:b}$ being ambiguous. Thus the model lacks the ability to recover from such mistakes during inference. To overcome this, we introduce a novel strategy in which we deliberately inject negative samples during post-training by substituting the previous query $\hat{Q}_b^{\text{prev}}$ in Eq. 4 with incorrect ones $Q_b^{\text{neg}}$. Crucially, when we perform negative sampling, we fall back to the pseudo ground truth query $\overline{Q_b^T}$ as the training label:

$$(\hat{Q}_b^T, \hat{Q}_b^{\text{prev}}) = \begin{cases} (\hat{Q}_b^T, \hat{Q}_b^{\text{prev}}), & \text{with probability } 0.9, \\ (\overline{Q_b^T}, Q_b^{\text{neg}}), & \text{with probability } 0.1. \end{cases} \quad (5)$$

This approach explicitly teaches the model to recover from errors in intermediate query prediction, thereby maintaining accuracy (see T. 6 for ablation) while achieving latency savings.

## 4 EXPERIMENT SETUP

### 4.1 EVALUATION BENCHMARKS

To rigorously evaluate our proposed approach, we construct comprehensive benchmark datasets featuring spoken queries paired with simulated tool interactions. We begin with the CRAG dataset (Yang et al., 2024), which form the basis for our spoken version of CRAG, which we term *AudioCRAG*. It consists of 2 distinct variants: ① **Audio CRAG Synthetic**: To generate spoken queries, we use our in-house TTS system. We then apply a rigorous filtering procedure which results in a high-quality set of 1,862 spoken queries, which we refer to as the *AudioCRAG-Synthetic* benchmark. ② **Audio CRAG Human**: To further enhance the realism and diversity of our evaluation, we introduce the AudioCRAG-Human benchmark, which consists of 618 human-recorded spoken queries. Further details on the construction of these benchmarks are provided in Sec. A.3. We follow the CRAG setup to incorporate both web and KG-based tools, and adopt its robust evaluation methodology, as described in Secs. A.4 and A.5. Additionally, we leverage a random subset of 16,000 questions from the text-based factual question answering dataset TriviaQA (Joshi et al., 2017) to post-train our speech-in, speech-out models, as detailed in Sec. A.7.

Table 2: Results on AudioCRAG-Synthetic for Qwen2.5-7B, OpusLM, and Kimi Audio comparing text vs. speech output across Closed Book, Open Book, and *Model-Triggered Stream RAG*.

| Setting | Ref length | Output | Model | Acc. |
|---------|-----------|--------|-------|------|
| Closed Book | 0 | Text | Qwen2.5-7B | 15.0 |
| | 0 | Speech | Qwen2.5-7B | 11.1 |
| | 0 | Text | OpusLM | 20.1 |
| | 0 | Speech | OpusLM | 18.4 |
| | 0 | Text | Kimi Audio | 24.2 |
| | 0 | Speech | Kimi Audio | 16.7 |
| Open Book | 23K | Text | Qwen2.5-7B | 39.6 |
| (S. 3.1) | 23K | Speech | Qwen2.5-7B | 26.3 |
| | 23K | Speech (self-cascade) | Qwen2.5-7B | 33.8 |
| | 15K | Text | OpusLM | 26.3 |
| | 15K | Speech | OpusLM | 0.0 |
| | 15K | Speech (self-cascade) | OpusLM | 21.2 |
| | 5K | Text | Kimi Audio | 45.8 |
| | 500 | Speech | Kimi Audio | 21.8 |
| Stream RAG | 23K | Text | Qwen2.5-7B | 39.8 |
| (S. 3.2.2) | 23K | Speech | Qwen2.5-7B | 34.2 |
| | 15K | Text | OpusLM | 29.7 |
| | 15K | Speech | OpusLM | 23.6 |

## 4.2 EVALUATED SOTA SPEECH-IN SPEECH-OUT MODELS

In this work, we present a comprehensive benchmark of three SOTA speech-in, speech-out conversational systems: Qwen-OMNI (Xu et al., 2025), Kimi-Audio (Ding et al., 2025) and OpusLM (Tian et al., 2025). We evaluate them under both tool-augmented and non-tool-augmented conditions. Further details about our experimental setup are provided in S. A.6.

We perform an ablation study (referred to as "Tool Integration" in T. 4) where we post-train the model on sequential query generation (i.e. $P(Q^T|Q)$ in S. 3.1) and output generation, to assess the impact of Stream RAG post-training versus standard post-training on final response generation. Additionally, we conduct an ablation study on open book setting (S. 3.1) using a self-cascade approach with a three-stage inference pipeline: (1) the audio question is used to generate a tool query $Q^T$ (corresponding to the "Query Generation" stage described in S. 3.1); (2) the audio question, and tool results are combined to produce the final text output $X^{\text{res}}$ by maximizing $P(X^{\text{res}}|Q, R)$; and (3) the audio question, and final text output are used to generate the final speech output $A$ by optimizing $P(A|Q, X^{\text{res}})$. Since we teacher-force the text output to obtain the final speech output in stage (3), this self-cascade approach can only be applied to a "thinker-talker" architecture (eq. Qwen-OMNI) or Chain-of-Thought (CoT) style architectures (eg. OpusLM). The motivation for this ablation is to investigate whether the inclusion of RAG references $R$ during the "Response Generation" stage (S. 3.1) affects the quality of the generated speech $A$.

## 5 RESULTS

### 5.1 IMPACT OF TOOL INTEGRATION AND STREAM RAG ON SOTA MODELS

Table 1 provides a comprehensive performance comparison of three models, Qwen2.5-7B, OpusLM, and Kimi Audio, evaluated across three settings: Closed Book, Open Book, and Stream RAG. All models are assessed on both the AudioCRAG-Synthetic (Syn.) and AudioCRAG-Human (Hum.). In the Closed Book setting, where models rely solely on their internal knowledge without access to external tools (reference length = 0), all models achieve accuracy scores below 20%. These results highlight the inherent limitations of closed-book approaches in handling complex queries. The Open Book setting, which provides models with access to external information, demonstrates the clear benefits of tool integration. Qwen2.5-7B and Kimi Audio's accuracy rises substantially, underscoring the value of leveraging external context. As expected, latency increases due to the additional overhead of retrieving information from external tools (See T. 3 for detailed latency analysis).

Table 3: First Token Latency breakdown, showing median (P50) and 90th percentile (P90) timings, for the Qwen2.5-7B on AudioCRAG-Synthetic.

| Model | Setting | P | Latency (sec) | | | |
|---|---|---|---|---|---|---|
| | | | Tool Use Latency | | Response Gen | Total |
| | | | Query Gen | Tool Results Gen | | |
| Qwen2.5-7B | Open Book (S. 3.1) | P50 | 0.59 | 2.78 | 2.52 | 5.90 |
| | | P90 | 0.85 | 4.90 | 3.25 | 9.00 |
| | Stream RAG (S. 3.2.2) | P50 | 0.59 | 2.20 | 2.52 | 5.32 |
| | | P90 | 0.85 | 4.37 | 3.25 | 8.47 |

Most notably, our post-training approach to build *Model-Triggered Stream RAG*, as described in S. 3.2.2, delivers significant advancements in both accuracy and efficiency. Qwen2.5-7B and OpusLM achieve significant accuracy improvements across both benchmarks. While the absolute accuracy scores may appear low, they are consistent with the evaluation results observed in the CRAG benchmark. Notably, Qwen2.5-7B with *Model-Triggered Stream RAG* achieves accuracy comparable to the open book performance of similarly sized LLMs reported in the original CRAG paper (i.e., 34.2 for Qwen2.5-7B[1] vs. 32.1 for LLAMA-3 8B Instruct in (Yang et al., 2024)). Importantly, although post-training is performed exclusively on the synthetic dataset, we observe consistent and even greater improvements on the human-spoken benchmark. This setting also introduces substantial latency savings compared to the Open Book configuration, with Qwen2.5-7B and OpusLM achieving 17.2% and 10.6% reductions in tool use latency (i.e. 1 – (Stream RAG Tool-Use Latency / Open-Book Tool-Use Latency) ) on the synthetic benchmark, and even greater savings (i.e. 53.4% and 35.3%) on the human benchmark [2]. These results demonstrate that our stream RAG approach not only advances the accuracy, but also optimizes response efficiency by enabling earlier and more effective prediction of tool queries.

## 5.2 ANALYSIS: MODALITY GAP BETWEEN SPEECH AND TEXT OUTPUT

Table 2 provides a comparative evaluation of the three SOTA models in generating either text or speech outputs from speech inputs, both with and without the integration of external tool results. In the absence of tool results (Ref length = 0), all models achieve higher accuracy when generating text outputs compared to speech outputs. The incorporation of tool results generally leads to improved text generation performance, with Kimi Audio achieving the highest accuracy. In contrast, the accuracy for speech output remains consistently lower across all models and conditions. The self-cascade approach, in which the model first generates an intermediate text response before producing the final speech output, provides moderate improvements in speech output accuracy for both Qwen2.5-7B and OpusLM. However, it still underperforms compared to the text-out baseline, primarily due to errors in accurately generating answers involving uncommon entity nouns. Overall, these findings underscore a persistent challenge in direct speech generation, particularly when tool results are integrated, as this appears to negatively impact the quality of generated speech responses. Our *Model-Triggered Stream RAG* demonstrates clear advantages: it maintains comparable performance for text output and delivers substantial improvements for speech output, even outperforming the self-cascade approach. These results underscore the effectiveness of post-training in overcoming challenges in direct speech generation in tool-augmented scenarios.

## 5.3 ANALYSIS OF LATENCY BOTTLENECKS IN TOOL-INTEGRATED SPEECH DIALOGUE

Table 3 provides a comprehensive breakdown of latency measurements for the Qwen2.5-7B speech-to-speech model, evaluated in both the Open Book setting and our proposed *Model-Triggered Stream RAG* setting on AudioCRAG-Synthetic. We report both median (P50) and 90th percentile (P90) values for each stage of the processing pipeline. The latency is decomposed into three main components: tool query generation, tool result generation, and speech response synthesis, with measurements provided for both the first token outputs (We also provide latency measurements for last

---

[1]With idealized text-in/text-out (i.e., perfect ASR and TTS), Qwen-OMNI achieves only modest increase to 38.9% accuracy, indicating that our method already operates close to the underlying LLM's accuracy ceiling.

[2]Our latency calculations on synthetic audio exclude end-point detection latency, which is required in all production systems. Since our stream RAG approach enables processing without waiting for end-point detection, including this factor would further amplify the observed latency savings, as observed in higher latency savings for human-spoken audio, where endpoint detection errors often introduce trailing silence.

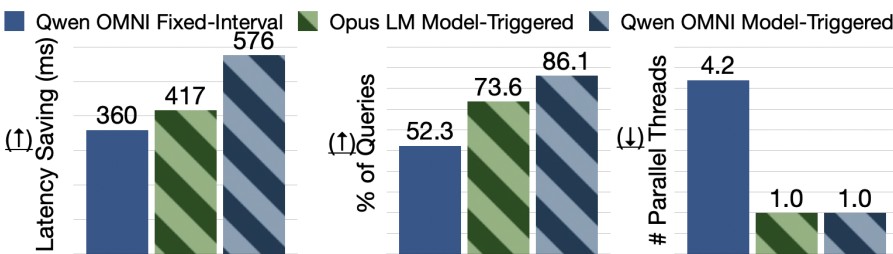

Figure 4: Latency savings by Stream RAG approaches (S. 3.2.)

Table 4: Comparison of speech-to-speech model under different post-training conditions

| Post-Training | Ref length | Post Train Data | Model | Acc. |
| --- | --- | --- | --- | --- |
| Tool Integration (S. 4.2) | 15K | 16K | OpusLM | 22.4 |
| Stream RAG (S. 3.2.2) | 15K | 16K | OpusLM | 23.6 |
| Tool Integration (S. 4.2) | 23K | 16K | Qwen2.5-7B | 34.9 |
| Stream RAG (S. 3.2.2) | 23K | 16K | Qwen2.5-7B | 34.2 |

token output in T. 9). For both tool query and tool result generation, it is assumed that all tools are accessed in parallel; thus, the reported latency corresponds to the maximum query or result generation time among all tools. The majority of this latency arises from leveraging external web pages, which introduces significant delays, most notably, increasing the first token latency by 2.3x in Open Book Setting. Notably, our *Model-Triggered Stream RAG* setting enables early generation of tool results, successfully reducing P50 first token latency by 9.8% and tool use latency by 17.2%. We additionally integrated vLLM for optimized inference leading to a **16.6%** reduction in P-50 first-token latency for Audio CRAG-Synthetic and an **57%** reduction for Audio CRAG-Human (see S. A.11), which are perceptually meaningful in the context of spoken dialogue (see S. A.12).

## 5.4 Ablation Study

**Stream RAG approaches**: Figure 4 highlights the substantial latency savings enabled by the *Model-Triggered Stream RAG* (S. 3.2.2), compared to the *Fixed-Interval Stream RAG* (S. 3.2.1). Three key metrics are evaluated: overall latency savings, the percentage of queries benefiting from reduced latency, and the number of parallel threads required. Even without any post-training, the *Fixed-Interval Stream RAG* approach already reduces tool usage latency (3.37s in T. 3) by 10.7% for Open Book Qwen-OMNI, demonstrating its flexibility and plug-and-play compatibility with any existing speech-in speech-out model. The *Model-Triggered Stream RAG* method, utilizing Qwen-OMNI, consistently delivers superior performance. It achieves greater average latency reductions and benefits a higher proportion of queries with improved response times. Notably, *Model-Triggered Stream RAG* require only a single parallel thread, representing a significant advancement in resource efficiency compared to the *Fixed-Interval Stream RAG* approach, which demands multiple parallel threads.

**Post-training Strategies**: Table 4 presents performance comparison under different post-training conditions. Incorporating streaming tool query generation during post-training (S. 3.2.2) results in comparable performance for both models. These results suggest that *Model-Triggered Stream RAG* can be integrated into post-training process without negatively impacting model performance.

## 6 Conclusion and Discussion

We introduced the first comprehensive approach to introduce streaming tool-query scheduling, i.e., issuing tool calls while the user is still speaking, into E2E speech-in, speech-out dialogue systems. Our *Model-Triggered Stream RAG* approach enhanced the model's ability to leverage retrieved information and autonomously decide when to trigger new tool queries, resulting in improved accuracy and responsiveness. Empirical evaluation on the newly introduced AudioCRAG benchmark demonstrated that tool integration can more than double factual question answering accuracy compared to closed-book models. Additionally, it achieved a 17% reduction in tool usage latency, thereby preserving natural conversational flow. Overall, our contributions advance the state of the art in spoken dialogue systems by enabling accurate, real-time, and tool-augmented voice assistants.

## 7 ETHICS STATEMENT

In this work, we propose an approach for integrating external tool usage directly into E2E speech-in, speech-out dialogue systems, and as such do not see any new ethical concerns arising as a result of our work. We are dedicated to upholding the highest standards of research ethics and reproducibility. All experiments utilize open-source models, ensuring no privacy violations, and we will release all code to the public to facilitate transparency and further research in the community. The AudioCRAG-Human dataset was commissioned and collected from consenting adult participants. All participants provided informed consent prior to their involvement in the study. Approximately 100 participants, all over the age of 18, were recruited by a third-party vendor and compensated for their participation. Personal identifying information was either obfuscated or not collected. The dataset is intended for evaluation purposes only and must not be used for training.

## 8 REPRODUCIBILITY STATEMENT

To ensure the reproducibility of our work, we provide comprehensive details throughout the paper and supplementary materials. The prompts used for query and response generation are included in S. A.9, while the evaluation prompts for the LLM-as-judge setup are detailed in S. A.10. Further information regarding tool usage and our experimental setup can be found in S. A.4 and A.6.The hyperparameters for our *Model-Triggered Stream RAG* post-training are summarized in Tables 10–12. In addition, we are committed to open science and will release our training code to support future research and development in this area.

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

# A APPENDIX

## A.1 BENCHMARKING TEXT DIALOGUE SYSTEMS FOR TOOL USAGE

Recent advances in benchmarking text-based dialogue systems for tool usage (Chen et al., 2023b; Ouyang et al., 2025; Cheng & Dou, 2025; Cohen et al., 2025; Xiong et al., 2024a; Vu et al., 2023; Xiong et al., 2024b; Peng et al., 2024; Su et al., 2025; Ni et al., 2025) have primarily focused on evaluating factual question answering and task completion within simulated environments. The CRAG benchmark (Yang et al., 2024) is a leading example, featuring 4,409 question-answer pairs and providing mock APIs for both web and knowledge graph (KG) search. CRAG supports a range of KG and web retrieval tasks, and highlights key challenges such as hallucinations in retrieval-augmented generation (RAG) and the importance of leveraging KGs and search ranking to improve factual accuracy. Evaluation is conducted automatically using two LLM judges. SimpleQA (Wei et al., 2024) is another widely adopted benchmark, designed to assess language models on short, fact-seeking questions. With 4,326 adversarially collected questions spanning diverse topics and a straightforward grading scheme based on single, indisputable answers, SimpleQA provides a robust testbed for factual accuracy. Moving beyond question answering, WebArena (Zhou et al., 2024) offers a simulated environment for evaluating dialogue agents on web-based tasks using fully functional websites, enabling assessment of more complex, action-oriented behaviors. While these benchmarks have significantly advanced the evaluation of tool-augmented dialogue systems, they remain largely limited to text-based interactions and do not fully address the unique challenges presented by speech-in speech-out systems.

## A.2 MULTIMODAL BENCHMARKS FOR TOOL USAGE

Recent benchmarks Mei et al. (2025); Yu et al. (2025); Luo et al. (2024) have extended tool-augmented dialogue evaluation to multimodal and longer-context scenarios. The m&m's benchmark (Ma et al., 2024) evaluates LLMs on multi-step, multi-modal tasks using a diverse set of 33 tools, including public APIs and multimodal models such as off-the-shelf automatic speech recognition (ASR) models, highlighting the potential for developing agents that leverage audio-based tools. CRAG_MM (Meta CRAG-MM Challenge Organizers, 2025) builds on the original CRAG benchmark by introducing visual question answering (QA) tasks that combine images and text-based queries, utilizing mock APIs for both image descriptions and web search. For video understanding and long-context reasoning, the Video Web Arena (Jang et al., 2025) benchmark evaluates multimodal agents on tasks involving 2,021 manually crafted tutorial videos. While these benchmarks advance the field by incorporating multimodal tools, they still do not evaluate systems in speech-in speech-out scenarios.

## A.3 AUDIO CRAG BENCHMARK

We begin with the CRAG dataset (Yang et al., 2024), which contains 2,706 text queries. Since these queries are not directly suitable for speech-based evaluation, we first identify those requiring adaptation before TTS conversion. Through careful manual inspection, we determine that queries containing dates or brackets benefit from rewriting to ensure naturalness and clarity in spoken form. In total, we identify 569 such queries and rewrite them using a large language model (LLAMA-4 Maverick). The resulting 569 rewritten queries, combined with the remaining original queries, form the basis for our spoken version of CRAG, which we term *AudioCRAG*. We follow the CRAG setup to incorporate web and KG-based tools and adopt its robust evaluation setup, as detailed in S. A.4 and A.5.

**Audio CRAG Synthetic**: To generate spoken queries, we process all 2,706 queries through our in-house TTS system. We then apply a rigorous filtering procedure to remove queries with intelligibility issues, specifically excluding any utterances for which Whisper Radford et al. (2023) hypotheses exhibit a non-zero word error rate. We also remove utterances with suboptimal audio quality, as determined by UTMOS (Saeki et al., 2022) scores below 3.5. This results in a high-quality set of 1,862 spoken queries, which we refer to as the *AudioCRAG-Synthetic* benchmark.

**Audio CRAG Human**: To further enhance the realism and diversity of our evaluation, we introduce the AudioCRAG-Human benchmark, which consists of 618 human-recorded spoken queries. These queries are recorded by a diverse pool of participants to capture natural variations in speech, accent,

and prosody. The inclusion of human-recorded audio enables a more comprehensive assessment of speech-in speech-out systems under real-world conditions, providing valuable insights into model robustness and generalization beyond synthetic speech. This benchmark serves as a critical resource for evaluating the effectiveness of tool integration in conversational AI systems.

## A.4 TOOL USAGE SETUP

To enable effective tool usage, we build upon the CRAG framework by integrating two complementary information sources: web search, which provides access to fresh and dynamic content, and a knowledge graph, which offers structured and reliable information. For web search, we aggregate all 100,000 documents from the CRAG corpus and employ a BGE-based re-ranker[3] Xiao et al. (2023) to index and retrieve the top 50 most relevant documents for each query. These documents are then segmented into chunks and re-ranked using the same BGE model based on their similarity to the query, ensuring highly contextually relevant retrieval. Meanwhile, queries to the knowledge graph are performed via a simulated API, adhering to the methodology established in CRAG [4].

Table 5: Results on AudioCRAG-Synthetic for Qwen2.5-7B, OpusLM, and Kimi Audio comparing text vs. speech output across Open Book setting showing average rates of accurate, hallucinated, and missing responses, as well as overall truthfulness scores for each system.

| Ref length | Output | Model | Score | Acc. | Halluc | Miss. |
|---|---|---|---|---|---|---|
| 0 | Text | Qwen2.5-7B | -13.1 | 15.0 | 28.1 | 56.9 |
| 0 | Speech | Qwen2.5-7B | -21.1 | 11.1 | 32.3 | 56.6 |
| 0 | Text | OpusLM | -44.3 | 20.1 | 64.3 | 15.6 |
| 0 | Speech | OpusLM | -47.9 | 18.4 | 66.2 | 15.4 |
| 0 | Text | Kimi Audio | -38.5 | 24.2 | 62.7 | 13.1 |
| 0 | Speech | Kimi Audio | -53.5 | 16.7 | 70.3 | 12.9 |

## A.5 EVALUATION SETTING

Similar to previous work (Yang et al., 2024), we employ model-based automatic evaluation. We use a three-way scoring system, assigning scores of 1, -1, and 0 for accurate, incorrect, and missing answers, respectively. The evaluation is conducted using the Llama 4-maverick LLM evaluator. For speech outputs, we first transcribe the audio using Whisper (Radford et al., 2023) before passing the transcriptions to the LLM evaluator. In this study, our primary focus is on enhancing system accuracy; therefore, we report average accuracy values in Tables 1 and 2. For additional context, we also provide the average rates of accurate, hallucinated, and missing responses, as well as overall truthfulness scores for each system in the Open Book Setting (see Table 5). Notably, our results indicate that Qwen-OMNI was less likely to generate hallucinated responses compared to OpusLM and Kimi Audio.

## A.6 EXPERIMENT SETUP OF SOTA SPEECH-IN SPEECH-OUT MODELS

**Qwen-OMNI** (Xu et al., 2025) is a end-to-end multimodal model that seamlessly integrates diverse input modalities—including text, images, audio, and video—and generates both text and natural speech responses in a real-time streaming fashion. It leverages an innovative Thinker-Talker architecture, where the Thinker module performs high-level reasoning to produce a text response, which is then used by the Talker module, conditioning on both the text and the Thinker's hidden representations, to generate streaming speech output.

**OpusLM** (Tian et al., 2025) is an open-source speech-in, speech-out model post-trained to directly answer complex semantic and factual questions from raw audio inputs, through Chain-of-Thought reasoning.

---

[3] https://huggingface.co/BAAI/bge-large-en-v1.5
[4] https://github.com/facebookresearch/CRAG/tree/main/mock_api

**Kimi Audio** (Ding et al., 2025) is a universal audio foundation model that unifies audio understanding, generation, and conversational abilities within a single framework. Pre-trained on over 13 million hours of diverse audio and text data, Kimi Audio achieves state-of-the-art performance across a wide range of audio benchmarks, including audio understanding and speech conversation tasks.

For tool-augmented scenarios, retrieval results are provided up to each model's maximum token limit ("Ref length" in Tables), maintaining a 2:1 ratio of web page to KG results. Specifically, we observe that Kimi-Audio is currently optimized for handling tool result references up to a certain length. When this limit is exceeded, an error arises during the audio detokenization process, specifically within the rotary embedding mechanism, highlighting an architectural constraint in processing longer input sequences or larger reference contexts. Addressing this limitation presents a valuable opportunity for future model enhancements.

Our evaluation encompasses both speech-in-text-out and speech-in-speech-out scenarios. For the Fixed-Interval Stream RAG setting (Section 3.2.1), intermediate tool queries are generated at consistent 1-second intervals. In the Model-Triggered Stream RAG setting, the model dynamically determines the need for a tool call after processing each 500ms block. This approach allows us to utilize a smaller chunk size, as only a single tool call thread is required for Model-Triggered Stream RAG, thereby enabling more efficient and responsive processing.

### A.7 POST TRAINING DATA PREPARATION

This subsection details the experimental setup for post-training the pretrained speech-in, speech-out model to significantly enhance its tool usage capabilities, as outlined in S. 3.2.2. We leverage a random subset of 16,000 questions from the text-based factual question answering dataset TriviaQA (Joshi et al., 2017), which contains 97,000 questions and 662,659 associated web documents. For *Model-Triggered Stream RAG*, we further compute word-level timestamps using a pre-trained ASR model, OWSM CTC v4 1B (Peng et al., 2025), enabling us to generate partial ASR transcripts $X_b^{\text{asr}}$ at 500 ms intervals. Note that if word occurs at boundary of block $b$, it is excluded from $X_b^{\text{asr}}$. For each partial transcript $X_b^{\text{asr}}$, we generate corresponding psuedo ground truth queries $\overline{Q_b^T}$ using LLAMA-4-Maverick to simulate incremental user input. To simulate realistic tool usage, we concatenate all documents and employ a web query reranker to retrieve the top 50 most relevant documents for each query. The text questions from this 16k subset are converted into discrete speech tokens using text-to-speech synthesis with the corresponding pretrained speech-in, speech-out model. Recognizing that TriviaQA answers are typically single named entities, we further transform the queries into a conversational style using LLAMA-4-Maverick, making them more suitable for dialogue-based evaluation.

### A.8 ABLATION RESULTS FOR NEGATIVE SAMPLING STRATEGY

Table 6: Ablation Results for Negative Sampling Strategy in *Model-Triggered Stream RAG* (S. 3.2.2).

| Scenario | Input | Ref Length | Post Train Data | Output | Model | Acc. |
|---|---|---|---|---|---|---|
| Open Book | Speech | 15K | 0 | Text | Qwen2.5-7B | 39.6 |
| Post-train (S. 3.2.2) | Speech | 15K | 16K | Text | Qwen2.5-7B | 39.8 |
| - Negative sampling | Speech | 15K | 16K | Text | Qwen2.5-7B | 36.5 |

This table presents the ablation results evaluating the impact of deliberately injecting negative samples during post-training (Eq. 5). The findings highlight that, without negative sampling, streaming tool query generation can reduce final accuracy in text output settings, primarily due to errors in final query generation, as detailed in S. 3.2.2. In contrast, our negative sampling approach significantly enhances the model's robustness, enabling it to recover from intermediate prediction errors. This leads to consistently high accuracy while also achieving notable latency reductions (Tab. 3).

Table 7: Example KG Queries, and Web Queries generated by Qwen-OMNI in Open Book setting

| ASR Transcript of Question $X^{\text{asr}}$ | Web Query $\hat{Q}^{\text{web}}$ | KG Query $\hat{Q}^{\text{KG}}$ |
|---|---|---|
| which of nolan greenwald's movies has achieved the highest level of box office success on a global scale? | Nolan Greenwald's highest-grossing movie | {'domain': 'movie', 'movie_name': "Nolan Greenwald's movies", 'movie_aspect': 'revenue'} |
| who has played drums for the red hot chili peppers? | Red Hot Chili Peppers drummers | {'domain': 'music', 'artist_name': 'Red Hot Chili Peppers', 'artist_aspect': 'member'} |
| what's the current stock price of tortoise midstream energy fund? | Tortoise Midstream Energy Fund stock price | {'domain': 'finance', 'market_identifier': 'Tortoise Midstream Energy Fund', 'metric': 'price', 'datetime': '02/28/2024'} |
| what was the volume of trading in cabot corporation's stock on the most recent day that dividends were distributed? | CABOT Corp stock trading volume on dividend distribution date | {'domain': 'finance', 'market_identifier': 'Cabot Corporation', 'metric': 'dividend', 'datetime': '02/28/2024'} |
| which movie won the academy award for best film in 2020? | 2020 Academy Award for Best Picture | {'domain': 'movie', 'movie_aspect': 'oscar_awards', 'year': 2020} |
| which teams have won against phoenix suns during 2022-12? | Teams that beat Phoenix Suns in December 2022 | {'domain': 'sports', 'sport_type': 'basketball', 'team': 'Phoenix Suns', 'datetime': '2022-12-15'} |

## A.9 PROMPTS USED FOR FACTUAL QA

### A.9.1 PROMPT IN CLOSED BOOK SETTING.

PROMPT = """" You are given an Audio Question and the time when it was asked in the Pacific Time Zone (PT), referred to as "Query Time". The query time is formatted as "mm/dd/yyyy, hh:mm:ss PT". Your task is to answer the question in as few words as possible.
Please follow these guidelines when formulating your answer:
1. If the question contains a false premise or assumption, answer "invalid question".
2. If you are uncertain or don't know the answer, respond with "I don't know".
### Question
{query}
### Query Time
{query_time}
### Answer
""""

### A.9.2 PROMPT IN OPEN BOOK / STREAM RAG SETTING.

PROMPT = """" You are given an Audio Question, References and the time when it was asked in the Pacific Time Zone (PT), referred to as "Query Time". The query time is formatted as "mm/dd/yyyy, hh:mm:ss PT". The references may or may not help answer the question. Your task is to answer the question in as few words as possible.
Please follow these guidelines when formulating your answer:
1. If the question contains a false premise or assumption, answer "invalid question".
2. If you are uncertain or don't know the answer, respond with "I don't know".
### Question
{query}

Table 8: Streaming Tool Queries generated by Qwen-OMNI after *Model-Triggered Stream RAG* post-training

| ASR Transcript of Question $X^{\mathbf{asr}}$ | Partial ASR Transcript $X_b^{\mathbf{asr}}$ | Web Query $\hat{Q}_b^{\mathbf{web}}$ | KG Query $\hat{Q}_b^{\mathbf{KG}}$ |
|---|---|---|---|
| Who founded Rare Beauty in 2019? | Who founded | Who founded what | {"domain": "other", "main_entity": "Who"} |
| | Who founded rare | Red Bull founder | {"domain": "other", "main_entity": "Red"} |
| | Who founded rare beauty | Who founded Rare Beauty | {"domain": "other", "main_entity": "Rare Beauty"} |
| | Who founded rare beauty in | NO_QUERY | NO_QUERY |
| | Who founded rare beauty in 2019? | NO_QUERY | NO_QUERY |
| How many jump shots did Darius Miles score with in the game on 2000-11-08? | how many | Number of queries on 03/27/2024 | {"domain": "other", "main_entity": "How many"} |
| | how many jump | Number of jump shots in basketball | {"domain": "other", "main_entity": "jump shot"} |
| | how many jump shots did | Derek Jeter jump shots count | {"domain": "sports", "sport_type": "other", "person": "Dairy", "datetime": "03/27/2024, 19:52:43 PT"} |
| | how many jump shots did Darius miles | Darius jump shots count | {"domain": "sports", "sport_type": "other", "person": "Darius", "datetime": "03/27/2024, 19:52:43 PT"} |
| | how many jump shots did Darius miles score | Darius Miles jump shots count | {"domain": "sports", "sport_type": "other", "person": "Darius Miles", "datetime": "03/27/2024, 19:52:43 PT"} |
| | how many jump shots did Darius miles score with in | Darius Miles jump shots scored | NO_QUERY |
| | how many jump shots did Darius miles score with in the game on | Darius Miles jump shots scored in game on 03/27/2024 | NO_QUERY |
| | how many jump shots did Darius miles score with in the game on November | Darius Miles jump shots scored in game on November 2024 | {"domain": "sports", "sport_type": "other", "person": "Darius Miles", "datetime": "November"} |
| | how many jump shots did Darius miles score with in the game on November 8 | Darius Miles jump shots scored on November 8 | {"domain": "sports", "sport_type": "other", "person": "Darius Miles", "datetime": "November 8"} |
| | how many jump shots did Darius miles score with in the game on November 8 | NO_QUERY | NO_QUERY |
| | how many jump shots did Darius miles score with in the game on November 8, 2000 | Darius Miles jump shots scored on November 8, 2000 | {"domain": "sports", "sport_type": "other", "person": "Darius Miles", "datetime": "November 8, 2000"} |

### Query Time
{query_time}
### References
# web
{web_results}

Table 9: Last-token Latency breakdown, showing median (P50) and 90th percentile (P90) timings, for the Qwen2.5-7B in Open Book Setting on AudioCRAG-Synthetic (First Token Latency=5.9 sec in T. 1).

| Model | Token | P | Latency (sec) | | | |
| | | | Tool Latency | | Response Gen | Total |
| | | | Query Gen | Tool Results Gen | | |
| Qwen2.5-7B | Last Token | P50 | 0.59 | 2.78 | 16.70 | 20.07 |
| | | P90 | 0.85 | 4.90 | 42.41 | 48.16 |

```
# knowledge graph
{kg_response}
### Answer
"""
```

### A.9.3 KG Query extraction in Open Book Setting.

PROMPT = """" You are an agent that only outputs JSON. You are given a Query and Query Time. Do the following:

1) Determine the domain the query is about. The domain should be one of the following: "finance", "sports", "music", "movie", "encyclopedia". If none of the domains apply, use "other". Use "domain" as the key in the result json.

2) Extract structured information from the query. Include different keys into the result json depending on the domains, and put them DIRECTLY in the result json. Here are the rules:

For 'encyclopedia' and 'other' queries, these are possible keys:
- 'main_entity': extract the main entity of the query.

For 'finance' queries, these are possible keys:
- 'market_identifier': stock identifiers including individual company names, stock symbols.
- 'metric': financial metrics that the query is asking about. This must be one of the following: 'price', 'dividend', 'P/E ratio', 'EPS', 'marketCap', and 'other'.
- 'datetime': time frame that the query asks about. When datetime is not explicitly mentioned, use 'Query Time' as default.

For 'movie' queries, these are possible keys:
- 'movie_name': name of the movie
- 'movie_aspect': if the query is about a movie, which movie aspect the query asks. This must be one of the following: 'budget', 'genres', 'original_language', 'original_title', 'release_date', 'revenue', 'title', 'cast', 'crew', 'rating', 'length'.
- 'person': person name related to moves
- 'person_aspect': if the query is about a person, which person aspect the query asks. This must be one of the following: 'acted_movies', 'directed_movies', 'oscar_awards', 'birthday'.
- 'year': if the query is about movies released in a specific year, extract the year

For 'music' queries, these are possible keys:
- 'artist_name': name of the artist
- 'artist_aspect': if the query is about an artist, extract the aspect of the artist. This must be one of the following: 'member', 'birth place', 'birth date', 'lifespan', 'artist work', 'grammy award count', 'grammy award date'.
- 'song_name': name of the song
- 'song_aspect': if the query is about a song, extract the aspect of the song. This must be one of the following: 'author', 'grammy award count', 'release country', 'release date'.

For 'sports' queries, these are possible keys:
- 'sport_type': one of 'basketball', 'soccer', 'other'
- 'tournament': NBA, World Cup, Olympic.

- 'team': teams that users are interested in.
- 'datetime': time frame that the user is interested in. When datetime is not explicitly mentioned, use 'Query Time' as default.

Return the results in a FLAT json.

*NEVER include ANY EXPLANATION or NOTE in the output, ONLY OUTPUT JSON!!!*
"""

### A.9.4 KG QUERY EXTRACTION IN STREAM RAG SETTING.

PROMPT = """" You are an agent that only outputs JSON. You are given an Audio Query, Previously generated JSON result ('Previous Result') and Query Time. Do the following:

1) Determine the domain the query is about. The domain should be one of the following: 'finance', 'sports', 'music', 'movie', 'encyclopedia'. If none of the domains apply, use 'other'. Use 'domain'as the key in the result json.

2) Extract structured information from the query. Include different keys into the result json depending on the domains, and put them DIRECTLY in the result json. Here are the rules:

For 'encyclopedia' and 'other' queries, these are possible keys: - 'main_entity': extract the main entity of the query.

For 'finance' queries, these are possible keys: - 'market_identifier': stock identifiers including individual company names, stock symbols. - 'metric': financial metrics that the query is asking about. This must be one of the following: 'price', 'dividend', 'P/E ratio', 'EPS', 'marketCap', and 'other'. - 'datetime': time frame that the query asks about. When datetime is not explicitly mentioned, use 'Query Time' as default.

For 'movie' queries, these are possible keys: - 'movie_name': name of the movie - 'movie_aspect': if the query is about a movie, which movie aspect the query asks. This must be one of the following: 'budget', 'genres', 'original_language', 'original_title', 'release_date', 'revenue', 'title', 'cast', 'crew', 'rating', 'length'. - 'person': person name related to moves - 'person_aspect': if the query is about a person, which person aspect the query asks. This must be one of the following: 'acted_movies', 'directed_movies', 'oscar_awards', 'birthday'. - 'year': if the query is about movies released in a specific year, extract the year

For 'music' queries, these are possible keys: - 'artist_name': name of the artist - 'artist_aspect': if the query is about an artist, extract the aspect of the artist. This must be one of the following: 'member', 'birth place', 'birth date', 'lifespan', 'artist work', 'grammy award count', 'grammy award date'. - 'song_name': name of the song - 'song_aspect': if the query is about a song, extract the aspect of the song. This must be one of the following: 'author', 'grammy award count', 'release country', 'release date'.

For 'sports' queries, these are possible keys: - 'sport_type': one of 'basketball', 'soccer', 'other' - 'tournament': NBA, World Cup, Olympic. - 'team': teams that users are interested in. - 'datetime': time frame that the user is interested in. When datetime is not explicitly mentioned, use 'Query Time' as default. Return the results in a FLAT json.
*NEVER include ANY EXPLANATION or NOTE in the output, ONLY OUTPUT JSON!!!*

3) Compare your newly generated result to the 'Previous Result'. **If your new result would be exactly the same as the 'Previous Result', output only NO_QUERY.** Return the results in a FLAT json.

Previous Result:
{prev_kg_query}
"""

### A.9.5 WEB QUERY EXTRACTION IN OPEN BOOK SETTING.

PROMPT = "''" You are given an Audio Query and Query Time. Your task is to generate a web query that can be used to retrieve relevant web pages. Rewrite the following query into a short and succinct form, focusing on the main topic or domain (e.g. finance, sports, music, movie, encyclopedia), key entities mentioned (e.g. people, organizations, locations), and specific aspects of those entities (e.g. performance metrics, relationships, events). Ensure the rewritten query is clear, concise, and easy to understand. Note that simply outputting the original query is not acceptable. You must rephrase the query to make it more concise and focused on the key information that will help retrieve relevant web pages.

For 'finance' queries, focus on: - Company names or stock symbols - Financial metrics (e.g. price, dividend, P/E ratio, EPS, marketCap) - Specific timeframes or events; if no timeframe is specified, use the Query Time as default

For 'sports' queries, focus on: - Sports Type (eg. basketball, soccer) - Teams, players - Statistics or performance metrics (e.g. scores, wins, losses) - Specific events or tournaments (eg. NBA, World Cup, Olympic) - Time frame that the user is interested in; if no timeframe is specified, use the Query Time as default

For 'music' queries, focus on: - Artist names or song titles - Specific aspects of artist (eg. band name, birth place, birth date, lifespan, artist work, grammy award count, grammy award date) - Specific aspects of song (eg. author, grammy award count, release country, release date) - Music genres or categories - Specific awards or recognition (e.g. Grammy Awards, Billboard)

For 'movie' queries, focus on: - Movie titles or celebrity names - Movie genres or other categories like budget, language, release_date, revenue, cast, crew, rating, length - Specific aspects of celebrity like acted_movies, directed_movies, oscar_awards, birthday - Specific awards or recognition (e.g. Oscars) For 'other' queries, focus on: - Main entity or topic - Specific aspects or attributes of the entity

When rewriting the query, ensure that it captures all important information from the original question that could impact the retrieval results. Do not omit any crucial details, such as specific dates, locations, or relationships between entities. Also, do not invent any new details on your own. If necessary, use the Query Time to provide context for the query. The goal is to create a concise and accurate query that effectively conveys the user's intent and retrieves relevant information. *NEVER include ANY EXPLANATION or NOTE in the output, ONLY OUTPUT THE REWRITTEN QUERY!!!* "''"

### A.9.6 WEB QUERY EXTRACTION IN STREAM RAG SETTING.

PROMPT = "'''You are given an Audio Query, previously generated Web query ('Previous Result') and Query Time.

Your task is to generate a web query that can be used to retrieve relevant web pages. Rewrite the following query into a short and succinct form, focusing on the main topic or domain (e.g. finance, sports, music, movie, encyclopedia), key entities mentioned (e.g. people, organizations, locations), and specific aspects of those entities (e.g. performance metrics, relationships, events). Ensure the rewritten query is clear, concise, and easy to understand.

Note that simply outputting the original query is not acceptable. You must rephrase the query to make it more concise and focused on the key information that will help retrieve relevant web pages.

For 'finance' queries, focus on: - Company names or stock symbols - Financial metrics (e.g. price, dividend, P/E ratio, EPS, marketCap) - Specific timeframes or events; if no timeframe is specified, use the Query Time as default

For 'sports' queries, focus on: - Sports Type (eg. basketball, soccer) - Teams, players - Statistics or performance metrics (e.g. scores, wins, losses) - Specific events or tournaments (eg. NBA, World Cup, Olympic) - Time frame that the user is interested in; if no timeframe is specified, use the Query Time as default

For 'music' queries, focus on: - Artist names or song titles - Specific aspects of artist (eg. band name, birth place, birth date, lifespan, artist work, grammy award count, grammy award date) -

Specific aspects of song (eg. author, grammy award count, release country, release date) - Music genres or categories - Specific awards or recognition (e.g. Grammy Awards, Billboard)

For 'movie' queries, focus on: - Movie titles or celebrity names - Movie genres or other categories like budget, language, release_date, revenue, cast, crew, rating, length - Specific aspects of celebrity like acted_movies, directed_movies, oscar_awards, birthday - Specific awards or recognition (e.g. Oscars)

For 'other' queries, focus on: - Main entity or topic - Specific aspects or attributes of the entity

When rewriting the query, ensure that it captures all important information from the original question that could impact the retrieval results. Do not omit any crucial details, such as specific dates, locations, or relationships between entities. Also, do not invent any new details on your own. If necessary, use the Query Time to provide context for the query. The goal is to create a concise and accurate query that effectively conveys the user's intent and retrieves relevant information. Now, compare the new web query to the previously generated web query ('Previous Result').

If the new query is similar enough to the previous web query (i.e., it effectively conveys the same user intent and would retrieve similar relevant information), output only *NO_QUERY*.

Previous Result:
{prev_web_query}
"""

## A.10   LLM AS JUDGE PROMPT

PROMPT="""
Assume you are a human expert in grading predictions given by a model. You are given a question and a model prediction. Judge if the prediction matches the ground truth answer by following these steps: 1: Take it as granted that the Ground Truth is always correct. 2: If the Prediction indicates it is not sure about the answer, "score" should be "0"; otherwise, go the next step. 3: If the Prediction exactly matches the Ground Truth, "score" is 1. 4: If the Prediction does not exactly match the Ground Truth, go through the following steps and likely give a score as 0. 5: If the Ground Truth is a number, "score" is 1 if and only if the Prediction gives a number that almost exactly matches the ground truth. 6: If the Prediction is self-contradictory, "score" must be 0. 7: If the prediction is not answering the question, "score" must be 0. 8: If the prediction is a concise and correct summary of the ground truth, "score" is 1. 9: If ground truth contains a set of items, prediction must contain exactly same items for the score to be 1. 10: Otherwise, "score" is 0.

Output a JSON blob with an "explanation" field explaining your answer as short as possible and an "score" field with value 1 or 0.

You should make the judgment based on provided examples. Examples: Question: "which company has higher eps, btu or cma?" Ground Truth: "cma" Prediction: "it is not possible to determine which company has a higher eps." Output: "score": 0, "explanation": "The prediction is not sure about the answer."

Question: "who is taller, a or b?" Ground Truth: "a" Prediction: "The answer is a. a is of height 1.75 m and b is of height 1.82 m. So b is taller." Output: "score": 0, "explanation": "The prediction is self-contradictory."

Question: "who among patricia place and lana clarkson is younger?" Ground Truth: "lana clarkson" Prediction: "lana clarkson (born april 5, 1962) is younger than patricia clarkson (born december 29, 1959)." Output: "score": 1, "explanation": "The prediction gives the correct answer."

Question: "did lakers win more games than warriors in 2024-4?" Ground Truth: "yes" Prediction: "yes, lakers won more games than warriors in 2023-2024 season." Output: "score": 0, "explanation": "The prediction is not answering the question."

Question: "what is the height of the building where citigroup is headquartered?" Ground Truth: "151 m" Prediction: "the height is 915 feet (279 m)." Output: "score": 0, "explanation": "The prediction, 151 m, does not match the ground truth, 279 m."

Question: "who were the last 3 prime ministers of uk?" Ground Truth: "rishi sunak, liz truss, and boris johnson" Prediction: "1. liz truss, 2. boris johnson, 3.theresa may" Output: "score": 0, "explanation": "The prediction does not contain item, rishi sunak, that is in the grount truth."

Question: "who authored the taming of the shrew?" Ground Truth: "william shakespeare" Prediction: "william shakespeare" Output: "score": 1, "explanation": "The prediction exactly matches the ground truth."

Question: "who authored The Taming of the Shrew?" Ground Truth: "william shakespeare" Prediction: "w shakespeare" Output: "score": 1, "explanation": "The prediction matches the ground truth as w is the abbreviation of william."

Question: "what is the state bird of california?" Ground Truth: "california quail" Prediction: "california valley quail" Output: "score": 1, "explanation": "The prediction matches the ground truth as california quail is also known as california valley quail."

Question: "what country is maharashtra metro rail corporation limited located in?" Ground truth: "india" Prediction: "indianapolis" Output: "score": 0, "explanation": "The prediction doesn't match the ground truth answers."

Question: "what was wolves's record in their last game in eng-premier league?" Ground Truth: "w" Prediction: "wolves's record in their last game in the eng-premier League was a 2-1 win over fulham." Output: "score": 1, "explanation": "The prediction correctly answers the result is a win."

Question: "how many seconds is 3 minutes 15 seconds?" Ground Truth: "195 seconds" Prediction: "3 minutes 15 seconds is 195 seconds." Output: "score": 1, "explanation": "The prediction matches ground truth as it correctly states the number of seconds asked in the question."

Question: "on which days did am distribute dividends in the last year?" Ground Truth: "2023-01-24, 2023-04-25, 2023-07-25, 2023-10-24" Prediction: "the company distributed dividends on october 24, 2023 and january 24, 2023." Output: "score": 0, "explanation": "The prediction partially matches the ground truth, but it is incomplete."

Question: "on which days did xxx distribute dividends in the last year?" Ground Truth: "2023-01-13, 2023-03-25, 2023-11-21" Prediction: "xxx distributed dividends on 1. 2023-01-13, 2. 2023-03-25, 3. 2023-10-21." Output: "score": 0, "explanation": "The prediction contains an item, 2023-10-21, which is not in ground truth."

Question: "who are current band members of maroon 5?" Ground Truth: "adam Levine, jesse carmichael, james valentine, matt flynn, pj morton, sam farrar" Prediction: "they are jesse carmichael, matt flynn, adam levine, pj morton, sam farrar, and james valentine." Output: "score": 1, "explanation": "The prediction exactly matches the ground truth."

Question: "which movies comprise the matrix franchise?" Ground Truth: "the matrix, the matrix reloaded, the matrix revolutions, the matrix resurrections" Prediction: "the matrix, the matrix reloaded, the matrix revolutions, the animatrix, and the matrix resurrections." Output: "score": 0, "explanation": "The prediction covers more items than what are given by the ground truth."

Question: "how deep is the deepest lake of new york?" Ground Truth: "618 ft" Prediction: "the deepest lake in new york is seneca lake, with a depth of 618.23 feet." Output: "score": 1, "explanation": "The prediction exactly matches the number in ground truth after rounding."

Question: "what is the closing price of meta yesterday?" Ground Truth: "$310.17" Prediction: "310.2" Output: "score": 1, "explanation": "The prediction exactly matches the number in ground truth after rounding."

Question: "what is the current market cap of appl?" Ground Truth: "2.81 trillion" Prediction: "2.667 trillion" Output: "score": 0, "explanation": "The prediction does not match the number in ground truth."

Question: "what is the current pe ratio of appl?" Ground Truth: "28.3" Prediction: "the current pe ratio of apple is 26.66" Output: "score": 0, "explanation": "The prediction does not match the number in ground truth."

Question: "how much is tesla's stock price down from its all-time high?" Ground Truth: "$221.83" Prediction: "209.52" Output: "score": 0, "explanation": "The prediction does not match the number in ground truth."

Question: "what is the length of amazon river?" Ground Truth: "over 4000 miles" Prediction: "the length of amazon river is 4,000 miles" Output: "score": 0, "explanation": "The prediction does not say Amazon River is longer than 4000 miles."

Question: "how many copies x were sold?" Ground Truth: "2 million." Prediction: "it is over 2 million" Output: "score": 0, "explanation": "The prediction does not match the ground truth."

Question: "what is the population of country x?" Ground Truth: "3,576,873" Prediction: "the population of country x is 3.3 million." Output: "score": 0, "explanation": "The prediction, 3.3 M, does not match the number, 3.6 M, in ground truth."

Question: "what is the current market value of stock x?" Ground Truth: "$2,237,578,268" Prediction: "$2.16 billion." Output: "score": 0, "explanation": "The prediction, 2.16 B, does not match the number, 2.2 B, in ground truth."

Question: "how old is michael jordan?" Ground Truth: "61" Prediction: "mj was born on February 17, 1963." Output: "score": 0, "explanation": "The prediction does not explicitly say the age."

Question: "how long did amy teach as a teacher?" Ground Truth: "24 years and 255 days" Prediction: "25 years." Output: "score": 0, "explanation": "The prediction does not exactly match the ground truth."

Question: "at what age did usher perform in the half time show of superbowl?" Ground Truth: "45 years old" Prediction: "usher has not performed at a super bowl halftime show." Output: "score": 0, "explanation": "The prediction does not match the ground truth."

Question: "what year did olivia rodrigo win the grammys best new artist award?" Ground Truth: "2022" Prediction: "sure! here's an answer in three words or fewer: 2022 grammy awards." Output: "score": 1, "explanation": "The prediction answers the correct number of the year."

Question: "how many times have zendaya and tom holland starred in a film together?" Ground Truth: "3" Prediction: "they have starred in 3 films together: spider-man: homecoming, spider-man: far from home, and spider-man: no way home." Output: "score": 1, "explanation": "The prediction gives the right number and it is reasonable."

Question: "who is younger, timothee chalamet or tom holland?" Ground Truth: "tom holland" Prediction: "timothée chalamet is younger than tom holland." Output: "score": 0, "explanation": "The prediction does not match the ground truth."

Question: "who had more number one hits on the us billboard, a or b?" Ground Truth: "a had more number one hits on the us billboard than b, with 20 number one hits compared to b's 15." Prediction: "a" Output: "score": 1, "explanation": "The prediction is a concise and correct summary of the ground truth."

Question: "what is xxx's birthdate?" Ground Truth: "1996-01-01." Prediction: "02/01/1996" Output: "score": 0, "explanation": "The prediction does not match the ground truth."

Question: "what was the worldwide box office haul for movie x?" Ground Truth: "101756123." Prediction: "102 million" Output: "score": 1, "explanation": "The prediction exactly matches the number in ground truth after rounding."

Question: "how much has spotify's user base increased by since 2020 in na?" Ground Truth: "spotify's user base increased by 34 million since 2020." Prediction: "spotify's north american user base increased from 36 million in 2020 to 85 million by 2021" Output: "score": 0, "explanation": "The prediction is not answering the question as it only gives the increase from 2020 to 2021."
""""

## A.11 VLLM INTEGRATION

Table 13 presents a detailed breakdown of first-token latency after integrating vLLM inference with Qwen2.5-7B on the AudioCRAG-Synthetic benchmark. We report both median (P50) and 90th-

Table 10: Post-training Parameters of OpusLM

| Parameter | Value |
|---|---|
| train_micro_batch_size_per_gpu | 1 |
| gradient_accumulation_steps | 2 |
| epochs | 2 |
| gradient_clipping | 1.0 |
| bf16 enabled | true |
| optimizer type | Adam |
| optimizer lr | 0.00001 |
| optimizer betas | [0.9, 0.95] |
| optimizer eps | 1e-8 |
| optimizer weight_decay | 3e-7 |
| optimizer adam_w_mode | true |
| scheduler type | WarmupDecayLR |
| scheduler warmup_type | linear |
| scheduler total_num_steps | 21534 |
| scheduler warmup_num_steps | 1077 |
| scheduler warmup_min_lr | 0 |
| scheduler warmup_max_lr | 0.00001 |

Table 11: Post-training Parameters of Qwen-OMNI Thinker

| Parameter | Value |
|---|---|
| bf16 | True |
| gradient_accumulation_steps | 4 |
| epochs | 1 |
| gradient_clipping | 1.0 |
| learning_rate | 7e-6 |
| lr_scheduler_type | cosine |
| warmup_ratio | 0.05 |
| per_device_train_batch_size | 1 |
| weight_decay | 0.01 |

Table 12: Post-training Parameters of Qwen-OMNI Talker

| Parameter | Value |
|---|---|
| bf16 | True |
| gradient_accumulation_steps | 4 |
| gradient_clipping | 1.0 |
| epochs | 2 |
| learning_rate | 5e-5 |
| per_device_train_batch_size | 1 |
| lr_scheduler_type | linear |
| warmup_ratio | 0.0 |
| weight_decay | 0.01 |

percentile (P90) timings, decomposed into tool-use latency—covering query generation and tool-results generation—and response-generation latency. As shown, vLLM integration yields a clear improvement in both Open-Book and Stream RAG settings. In the Stream RAG configuration, the P50 total first-token latency decreases from 3.50 s to 2.92 s (16.6% relative reduction), driven primarily by reduced tool-results generation time (2.78 s → 2.20 s). Complementing these results, we also observed substantial gains on AudioCRAG-Human, where vLLM enables a 57% reduction in median first-token latency (3.16 s → 1.36 s).

Table 13: First Token Latency breakdown, showing median (P50) and 90th percentile (P90) timings, for the Qwen2.5-7B on AudioCRAG-Synthetic after integrating with vLLM.

| Model | Setting | P | Latency (sec) | | Response Gen | Total |
|---|---|---|---|---|---|---|
| | | | Tool Use Latency | | | |
| | | | Query Gen | Tool Results Gen | | |
| Qwen2.5-7B | Open Book (S. 3.1) | P50 | 0.05 | 2.78 | 0.67 | 3.50 |
| | | P90 | 0.07 | 4.90 | 0.86 | 5.83 |
| | Stream RAG (S. 3.2.2) | P50 | 0.05 | 2.20 | 0.67 | 2.92 |
| | | P90 | 0.07 | 4.37 | 0.86 | 5.30 |

Table 14: Performance comparison of Exact Match (EM) accuracy and Latency Savings (i.e. (Open-Book First Token Latency) – (Stream RAG First-Token Latency)) for Qwen2.5-7B, across Stream RAG (*Model-Triggered Stream RAG*) setting. We compared our approach with WavRAG (Chen et al., 2025) using GPT-4o and QwenAudio base models for response generation. * Unlike WavRAG-CoT, we do not use Chain-of-Thought or Self-Consistency during response generation. Also note that StreamRAG retrieves text documents using generated web query where WavRAG baselines retrieve the spoken version of the corresponding document.

| Setting | # Docs | Modality of Doc | Model | EM | Latency Savings (s) |
|---|---|---|---|---|---|
| WavRAG | 3 | Audio | GPT-4o | 40.1 | 0 |
| (Chen et al., 2025) | 3 | Audio | QwenAudio | 30.6 | 0 |
| WavRAG-CoT* | 3 | Audio | GPT-4o | 49.2 | 0 |
| (Chen et al., 2025) | 3 | Audio | QwenAudio | 34.0 | 0 |
| Stream RAG | 3 | Text | Qwen2.5-7B | **51.2** | **1.84** |

## A.12 HUMAN PERCEPTION OF LATENCY SAVINGS

Prior work on human conversational timing (Jacoby et al., 2024; Stivers et al., 2009) shows that the average turn-taking gap in natural human dialogue is approximately 239 ms, with delays beyond 500 ms perceived as unnatural. Complementary industry analyses (Vapi, 2025) similarly report that speech latency exceeding 500 ms begins to degrade user experience, causing users to interrupt or disengage. In this context, our observed latency reductions, particularly a 1.8 s improvement (T. 1) for human-spoken audio, represent a substantial enhancement relative to perceptual thresholds in human conversation. Moreover, our integration of efficient inference backends (e.g., vLLM for Qwen-OMNI) achieves P-50 latency near 670 ms, comparable to production-quality voice AI systems that target sub-second responsiveness. These results indicate that our latency reductions are perceptually significant, contributing directly to smoother conversational turn-taking and more natural user experience in speech-to-speech dialogue systems.

## A.13 RESULTS ON SLUE-SQA DATASET

T. 14 reports the Exact Match (EM) accuracy and latency savings of StreamRAG on the SLUE-SQA (Shon et al., 2023) benchmark, compared against prior RAG baselines such as WavRAG (Chen et al., 2025). The results show that StreamRAG performs competitively with existing RAG approaches, achieving a 51.2 EM and outperforming all WavRAG variants[5]. More importantly, StreamRAG delivers a 1.84-second reduction in first-token latency, demonstrating that the proposed strategy yields substantial efficiency gains without compromising accuracy. To address concerns regarding dependence on synthetic post-training data, we emphasize that these improvements are observed on an entirely different benchmark (SLUE-SQA) without any additional fine-tuning, indicating that the method generalizes beyond the Audio CRAG setting. This cross-benchmark consistency suggests that StreamRAG's accuracy and latency benefits are not specific to the CRAG benchmark but instead reflect robust generalization across datasets and model architectures.

---

[5]Note that StreamRAG retrieves text documents, whereas WavRAG retrieves spoken versions of the documents; thus the results are not directly comparable.

Table 15: Performance comparison of ROUGE and Latency Savings (defined as (Open-Book First-Token Latency) – (Stream RAG First-Token Latency)) for Qwen2.5-7B in the speech-in/speech-out setting. Results are reported for the second turn of two-turn interactions in the CORAL dataset, evaluating the effectiveness of Stream RAG in multi-turn retrieval scenarios.

| Setting | # Docs | Model | ROUGE-L | Latency Savings (s) |
|---|---|---|---|---|
| Open Book | 10 | Qwen2.5-7B | 10.7 | 0 |
| Stream RAG | 10 | Qwen2.5-7B | **11.3** | 2.27 |

### A.14 EVALUATING ROBUSTNESS TO MISFIRED EARLY TOOL CALLS

To further evaluate the robustness of the proposed Stream RAG framework in ambiguous scenarios, we conduct an additional analysis on the SLUE-SQA dataset. In particular, we simulate a challenging failure mode by negative sampling previous query $\hat{Q}_b^{\text{prev}}$ from SLUE-SQA's candidate query set[6], exposing the system to misleading intermediate information. We define the gold query as the final query produced by Stream RAG under the standard setting without negative sampling, and consider a perturbed prediction correct if it either (i) exactly matches this gold query or (ii) retrieves the same top-$k$ document with $k <= 5$. Under this evaluation, Stream RAG achieves an accuracy of 65.4%, demonstrating that even when confronted with intentionally misleading intermediate queries, the model is able to recover the correct final retrieval target in the majority of cases. This result highlights the framework's inherent robustness to early misfires and confirms its ability to revise and correct tool calls as more of the user's speech becomes available.

### A.15 EVALUATING PERFORMANCE IN MULTI TURN SCENARIOS

T. 15 reports a comparison of ROUGE performance and latency characteristics between the Open-Book and Stream RAG settings for Qwen2.5-7B in the speech-in/speech-out setup on multi-turn queries. To construct this evaluation, we converted the first two turns of each example in the CORAL dataset (Cheng et al., 2025) into spoken form using high-fidelity TTS[7] (Hayashi et al., 2020) and employed Whisper to transcribe the audio, manually verifying that no intelligibility or transcription errors were introduced. From this corpus, we selected 93 two-turn interactions with perfect transcription fidelity as a targeted stress test of our streaming RAG approach under multi-turn conditions. We then conducted a controlled comparison between Stream RAG, operating only on the first and second spoken turns, and an Open-Book baseline using the identical dialogue history, with both conditions constrained to 25-word generations to ensure fair evaluation. Notably, Qwen2.5-7B achieves a ROUGE score of 13.1 on the full CORAL test set (Cheng et al., 2025) in the text-in/text-out setting, indicating that the absolute ROUGE values observed here are reasonable given the additional challenges of speech processing. Consistent with the single-turn findings (T. 14, 1), Stream RAG continued to deliver substantial first-token latency reductions while also achieving higher ROUGE scores on the second turn, demonstrating that the benefits of streaming retrieval extend robustly to multi-turn contexts where user intent depends on preceding dialogue turns.

---

[6]The candidate query set consists of all final queries generated by the Stream RAG approach on the SLUE-SQA dataset.

[7]https://huggingface.co/espnet/kan-bayashi_libritts_xvector_vits

