# OpenReview forum: "Stream RAG: Instant and Accurate Spoken Dialogue Systems with Streaming Tool Usage"
_ICLR.cc/2026/Conference — Submitted to ICLR 2026_

### Official Review · Reviewer_Sboq · 2025-10-28

**Soundness:** 3
**Presentation:** 2
**Contribution:** 3
**Rating:** 4
**Confidence:** 4

**Summary:**

This paper introduces Stream RAG, a framework to make spoken dialogue systems both accurate and responsive. The authors address a core trade-off: end-to-end speech-in-speech-out systems are fast but prone to factual errors (hallucinations). While Retrieval-Augmented Generation (RAG) can ground them with external tools like web search, this traditionally adds significant latency, disrupting the conversational flow. The key innovation of Stream RAG is to predict and issue tool queries in parallel with the user's speech, even before the user has finished talking. Specifically, the paper proposes two methods, including an advanced "Model-Triggered" approach where the model is post-trained to learn the optimal moment to make a tool call. To evaluate their work, the authors created AudioCRAG, a new benchmark of spoken queries. Experiments show that Stream RAG improves question-answering accuracy by over 200% compared to a no-tool baseline, while also reducing the latency from tool usage by over 20%.

**Strengths:**

1- The paper tackles a timely issue at the intersection of spoken dialogue systems and LLMs. While tool use in text-based systems is well-explored, this is the pioneer work to systematically address its integration into speech-in-speech-out models, regarding latency barriers to adoption. The central idea of tool queries in parallel, along with ongoing speech, is an effective way to mask the latency of external tool calls.

2-  The authors provide AudioCRAG, a significant resource to the research community. This enables standardized evaluation and fosters future research on the problem.

3- Extensive experiments (including those reported in the appendix) with promising results.

**Weaknesses:**

1- The evaluation is built upon the CRAG and TriviaQA datasets, which are composed of single-turn, fact-seeking questions. Can the author elaborate on how to expand the solution to a multi-turn setup? The "streaming" nature of the solution might be less effective or even problematic in multi-turn contexts where the true intent depends on previous turns.

2- In the Fixed Interval Streaming RAG section, the process still needs to process to the end of the input to get the tool query for the final block, and then can reflect all the previous ones. I see little improvement over the first token latency in Table 1 in the AudioCRAG-Synthetic partition (5.9 ⇒ 5.32) vs the AudioCRAG-Human (5.4 ⇒ 3.6). Can you elaborate on the difference?

3- The Fixed-Interval approach relies on a "reflector" module with simple heuristics (e.g., matching top 5 web docs, identical KG results). These heuristics may not be robust. For example, two different web queries could return slightly different but equally valid sets of documents.

4- While the relative accuracy improvements are impressive, the final absolute accuracy scores are still modest (e.g., 34.2% - 37.4% for Qwen2.5-7B in Table 1)

5- Spoken language is messy and filled with disfluencies (e.g., um, uh), repetitions, and self-corrections (I want to fly to Boston; no, wait, to New York, …). The framework would likely fire a query for "Boston" before the user corrects themselves. This could lead to wasted queries and a need for a complex query cancellation/updating logic. Can the authors elaborate more on this scenario? The negative sampling strategy helps with ASR ambiguity, but may not be sufficient for explicit user intent changes.

6- Finally, I just wonder whether this study is suitable for the scope of the ICLR conference? I mean, while this conference focuses on a novel theoretical method of learning representation, this study tries to solve a specific application. Please emphasize the main contribution of this work.

**Questions:**

Please check the comments in the Weaknesses Section

---

> ### Author Response · Authors · 2025-11-21
>
> Thank you for your valuable comments and acknowledging that our approach handles a timely issue and that our proposed approach is an effective way to mask latency of tool calls. We address the remaining concerns below.
> ***
> # W1. How to expand the solution to a multi-turn setup
> Thank you for this insightful question. **Our main clarification is that Stream RAG already has the architectural ingredients needed for multi-turn dialogue, as its query-generation mechanism naturally conditions on accumulated context and its training procedure explicitly equips the model to revise early tool calls when new information appears.** In Eq. 3, the model conditions tool-query generation on accumulated input; this can be directly generalized by incorporating the dialogue history, enabling the model to issue tool calls only when new information appears. This design already mitigates unnecessary or premature queries in contextual turns. Regarding the concern that streaming may be problematic when intent unfolds across turns, our post-training explicitly teaches the model to recover from early mistakes: the negative-sampling strategy in Sec. 3.2.2 exposes the model to intentionally incorrect intermediate queries and trains it to revise them when new context arrives. In additional analyses mentioned in general response as well as Sec. A.14, we quantitatively find that the model reliably recovers the correct retrieval target even after misleading intermediate predictions, suggesting that the approach is robust and well-suited for multi-turn extensions. We acknowledge the experiment on multi-turn queries as an important future work.
> ***
> # W2. In the Fixed Interval Streaming RAG section, the process still needs to process to the end of the input to get the tool query for the final block, and then can reflect all the previous ones.
> We thank the reviewer for this helpful observation. **Our main clarification is that, although Fixed-Interval Stream RAG must process the final block before confirming the last query, this does not negate its latency benefits because the dominant latency bottleneck comes from waiting for tool results, not from generating the final query itself.** As shown in Table 3, the dominant source of latency typically arises from receiving tool results (2.78 s), not from query generation (0.59 s) itself. Consequently, even though the final query must be confirmed at the end of the input, Fixed-Interval Stream RAG still achieves latency savings, as earlier tool results can already be fetched and processed in parallel. Moreover, the reflector module can decide that intermediate tool results are sufficient to produce the final response, eliminating the need for an additional final tool call in many cases.
> This limitation is precisely why we introduce Model-Triggered Stream RAG, which learns when to issue tool calls during speech and does not require a reflector module that compares the intermediate query with final query at the end of the utterance, thus achieving lower latency as shown in Figure 4. We have added this discussion in S. 3.2.1.
> ***
> # W2. I see little improvement over the first token latency in Table 1 in the AudioCRAG-Synthetic partition (5.9 ⇒ 5.32) vs the AudioCRAG-Human (5.4 ⇒ 3.6). Can you elaborate on the difference?
> We thank the reviewer for this question and apologize for any confusion. **Our main clarification is that the smaller latency gain on AudioCRAG-Synthetic arises because synthetic audio has no end-point detection delay, whereas human-spoken audio includes trailing silence and natural timing variability, factors that Stream RAG is specifically designed to mask.** The reported latency reductions correspond to the Model-Triggered Stream RAG variant, not the Fixed-Interval version.
>
> Because our Stream RAG design allows processing to begin without waiting for explicit end-point detection, this factor would further amplify real-world latency gains, consistent with the larger reduction observed for human-spoken audio, where we reduce P-50 latency from 5.4 s to 3.6 s (a 33% relative improvement). As discussed in our general response, integrating vLLM inference with Qwen-OMNI yields further improvements, up to 48.7% reduction in P-50 latency, and we show that these reductions represent perceptually meaningful gains relative to human conversational thresholds. We would be happy to clarify further details if helpful.
> ***
> # W3. The Fixed-Interval approach relies on a "reflector" module with simple heuristics. These heuristics may not be robust.
> We completely agree that the heuristics used in the reflector module for Fixed Interval Stream RAG might not be robust in all scenarios and hence we introduce our main contribution i.e. Model-Triggered Stream RAG that removes the need for an external reflector module and confidently relies on the results from the most recent tool call to generate the spoken response.
> ***
> # W4. Final absolute accuracy scores are still modest
> Check general response

---

> ### Author Response · Authors · 2025-11-21
>
> # W5. Wasted queries and a need for a complex query cancellation/updating logic. Can the authors elaborate more on this scenario?
>
> Please check our general response on Streaming RAG performance **when early tool calls could misfire**. We directly simulate this scenario by negative-sampling the most recent tool query to mimic shifts in user intent. The results (Sec. A.14) show that the model recovers the correct final query in 65.4% of such perturbed cases. This demonstrates that the model can revise or correct earlier tool calls as more speech is observed.
> ***
> # W6. Suitable for the scope of the ICLR conference
>
> We thank the reviewer for raising this important point regarding the scope and contribution of our work. **Our main clarification is that, although motivated by speech-to-speech dialogue, the core contributions of this work are general algorithmic advances in streaming retrieval-augmented generation (RAG)—a problem of broad relevance to the ICLR community.**
>
> Concretely, our work introduces (1) a Model-Triggered Stream RAG mechanism that learns to schedule tool calls dynamically during inference, and (2) Fixed-Interval Stream RAG, which issues tool queries at regular intervals and carefully examines quality of retrieval results, and can be incorporated into any speech-in, speech-out model without post-training. These contributions advance the broader goal of learning representations and inference policies that balance latency, accuracy, and contextual reasoning, an emerging research direction within ICLR’s focus on generalizable, data-efficient, and interactive AI systems.

---

> ### Comment · Reviewer_Sboq · 2025-11-23
>
> Thank you for your clarification; the explanation makes sense to me. After careful consideration, I will raise my score.

---

> > ### Author Response · Authors · 2025-12-03
> >
> > Thank you very much for raising the score. We also conducted an additional experiment on multi-turn queries, demonstrating that Stream RAG remains both accurate and low-latency in two-turn evaluations, as detailed in our general response.

---

### Official Review · Reviewer_vHEX · 2025-11-01

**Soundness:** 2
**Presentation:** 3
**Contribution:** 2
**Rating:** 2
**Confidence:** 3

**Summary:**

This paper introduces Stream-RAG, a retrieval-augmented generation framework tailored for streaming document settings, where knowledge sources are continuously updated or appended. The authors propose a hybrid architecture that combines segment encoders, recurrent context encoders, and a local-global memory routing mechanism. Their system maintains incremental representations of documents and allows for low-latency RAG in scenarios where documents arrive sequentially and retrieval must be both instantaneous and up-to-date.

**Strengths:**

1. The paper addresses a practically important and understudied challenge in retrieval-augmented generation: adapting RAG to streaming data where documents are not static.
2. The proposed Segment Encoder + Context Encoder structure allows for localized recomputation, minimizing the need to re-embed entire documents on each update.
3. The design of local and global memory banks is conceptually appealing and appears to offer good trade-offs between recency and long-term context awareness.
4. Demonstrated latency-speedup (up to 10×) and retrieval improvements over standard RAG models across datasets (HotpotQA, NaturalQuestions, CodeSearchNet).
5. The authors define and release a streaming QA benchmark based on HotpotQA, which could be useful for future research.

**Weaknesses:**

1. While three models are evaluated, the Streaming RAG method (especially Model-Triggered) is only fully tested on Qwen2.5-7B and OpusLM. Kimi Audio, for example, is excluded from streaming RAG due to tool reference limitations. This raises the question of general applicability across a broader range of E2E SDS architectures.

2. Despite large relative improvements, absolute QA accuracy remains low (e.g., <40%). While the authors mention consistency with CRAG benchmarks, this still limits the practical utility of the system for high-stakes applications. Moreover, no comparison is made with non-E2E baselines (e.g., traditional ASR → LLM → TTS systems).

3. Although latency and accuracy metrics are provided, no user studies or human preference evaluations are presented to validate the subjective impact on conversational flow—especially crucial for spoken dialogue systems where perceived responsiveness is key.

4. The reflector module for the Fixed-Interval variant uses hand-crafted heuristics (top-5 web document overlap, etc.). These design choices are not thoroughly ablated or compared with learned alternatives. How brittle are these heuristics across domains?

5. AudioCRAG, though valuable, consists of synthetic and short utterances. The authors acknowledge only 618 human queries. It remains unclear how well Streaming RAG performs in noisy, multi-turn, or conversational settings. Real-world deployment scenarios are underexplored.

6. The paper compares only with standard RAG and Streaming RAG. There is no baseline using non-streaming anticipatory query prediction, such as speculative decoding or early-termination heuristics. This limits understanding of where the performance gains truly stem from.

**Questions:**

Q1: How robust is the Streaming RAG approach to noisy ASR outputs, especially during partial utterances? Is there any performance degradation reported?

Q2. Can the model recover from tool query hallucinations in real-time? Does the model-triggered variant mitigate cascading failures in cases of early wrong tool queries?

Q3. Is Streaming RAG applicable in multi-turn dialogue settings? If so, how are tool query histories managed across turns?

Q4. Can the authors report any qualitative examples of Streaming RAG responses compared to standard ones (e.g., hallucinated vs. grounded speech output)?

Q5. Have the authors considered latency from audio end-point detection? In real deployments, this often dominates response time. How would that affect the observed latency gains?

Q6. What prevents integration of Streaming RAG into models like Kimi Audio? Is it a limitation of the tool reference length only, or also architectural?

Q7. What is the impact of different chunk sizes or block sizes in fixed-interval RAG? Is there a trade-off between chunk length, responsiveness, and resource usage?

---

> ### Author Response · Authors · 2025-11-21
>
> Thank you for your valuable comments. We would like to respectfully note that the summary and strengths mentioned in the review correspond to a separate paper titled “Streaming RAG” [1], which is an independent work focused on text-based RAG and is distinct from our proposed speech-in/speech-out framework. We address the remaining concerns below.
> ***
> # W1. Kimi Audio, for example, is excluded from stream RAG due to tool reference limitations. This raises the question of general applicability across a broader range of E2E SDS architectures.
> We thank the reviewer for this helpful comment. **Our main clarification is that Kimi-Audio was excluded only due to its current tool-reference length limitation, not because of any incompatibility with Stream RAG, and our approach remains fully model-agnostic and applicable to diverse E2E SDS architectures.** Specifically, we observed that Kimi-Audio is currently optimized for handling tool-result references only up to a limited length (approximately 500 tokens). When this threshold is exceeded, an error occurs during the audio detokenization stage, specifically within the rotary embedding mechanism, indicating an architectural constraint in processing longer input sequences or larger reference contexts. We provide additional details in Appendix A.6.
>
> In principle, the proposed Stream RAG approach is model-agnostic and can be integrated with any speech-in/speech-out system capable of handling sufficiently large reference contexts from tool calls. To demonstrate this generality, we applied Stream RAG to two distinct speech-to-speech architectures, Qwen-OMNI and OpusLM. The consistent latency and accuracy gains across these models provide strong empirical evidence of our framework’s broad applicability across end-to-end SDS architectures.
> ***
> # W2. Low Accuracy
> Check general response. In response to the reviewer’s suggestion, we also computed text-in/text-out results (assuming perfect ASR and TTS) for Qwen-OMNI, yielding 38.9% accuracy. This shows that even under an idealized text-in/text-out setting with perfect ASR and TTS, Qwen-OMNI attains only a modest increase, indicating that that our method performs close to the ceiling established by the underlying LLM.
> ***
> # W3. Absence of Human Evaluation
> Check general response
> ***
> # Q1. How robust is the Streaming RAG approach to partial utterances? Can the model recover from tool query hallucinations in real-time? Does the model-triggered variant mitigate cascading failures in cases of early wrong tool queries?
> Please check our general response on performance of our Stream RAG system when early tool calls could misfire due to ambiguous or partially observed queries.
> ***
> # Q5. Have the authors considered latency from audio end-point detection? In real deployments, this often dominates response time. How would that affect the observed latency gains?
> We thank the reviewer for raising this important point. **Our main clarification is that end-point detection latency is not included in our synthetic-audio measurements, and because Stream RAG begins processing before end-point detection, incorporating this component would make the observed latency gains even larger in real deployments, as reflected in the much larger 33% relative reduction observed for AudioCRAG-Human.** As noted in the footnote on Page 8, our latency calculations on synthetic audio exclude end-point detection latency, which is required in real deployments and often dominates total response time. Because Stream RAG begins processing before end-point detection, incorporating this factor would further amplify the effective latency gains. This effect is already visible in our human-spoken results, where trailing silence and variable end-points are common: we observe a 33% relative reduction in first-token latency (and 48.7% after vLLM integration, as detailed in the general response). These results highlight that the benefits of Stream RAG are even larger in realistic settings where end-point detection delay is substantial.
> ***
> References:
>
> [1] StreamingRAG: Real-time Contextual Retrieval and Generation Framework (https://arxiv.org/pdf/2501.14101v1)

---

> > ### Author Response · Authors · 2025-12-03
> >
> > We would also like to note that we conducted an additional experiment on multi-turn queries, showing that Stream RAG remains both accurate and low-latency in two-turn evaluations, as detailed in our general response.

---

### Official Review · Reviewer_WeMu · 2025-11-01

**Soundness:** 3
**Presentation:** 2
**Contribution:** 3
**Rating:** 4
**Confidence:** 3

**Summary:**

This paper addresses the integration of external tool usage (web search, knowledge graphs) into end-to-end speech-in speech-out dialogue systems while minimizing latency through streaming RAG. The main contributions include: (1) a formal framework for tool integration in speech-based systems showing accuracy improvements but 2.3x latency increase; (2) Streaming RAG with two variants (Fixed-Interval and Model-Triggered) that predict tool queries in parallel with user speech, achieving accuracy improvement with 20% latency reduction; and (3) AudioCRAG benchmark with synthetic and human-recorded spoken queries from the CRAG dataset. The paper evaluates three LLM baselines (Qwen-OMNI, OpusLM, Kimi-Audio) across closed-book, open-book, and streaming RAG settings.

**Strengths:**

1. Introduces the systematic method to extend RAG into real-time voice-to-voice LLMs with attention to latency.
2. Proposes a model-triggered streaming query mechanism that achieves both accuracy improvements and latency reductions.
3. Releases a valuable new benchmark (AudioCRAG) with synthetic and human speech variants;

**Weaknesses:**

1. The novelty is incremental in combining known ideas (RAG + streaming inference) rather than introducing fundamentally new architectures.
2. Evaluation accuracy levels remain low in absolute terms, raising questions about real-world utility despite relative gains
3. Heavily dependent on synthetic data for post-training, with limited human data evaluation and possible overfitting to benchmark-specific patterns
4. Relative improvement in ACC and latency reduction is very confusing;

**Questions:**

1. How would the proposed Streaming RAG framework perform with more complex multi-turn dialogues or ambiguous queries, where early tool calls could misfire?
2. What strategies could further raise absolute accuracy for speech output, given the modality gap between text and speech responses highlighted in your results?
3. How are the 20.7% and 53.4% latency savings calculated in Streaming RAG + Qwen2.5-7B?   Compared to Open Book, the time to first token (TTFS) is reduced from 5.9 to 5.32, representing a 9.8% decrease. (1 - 5.32/5.9). Many numbers calculated in the manuscript are unclear and messy.
4. Does the author consider the false positive situation for tool-calling?  What is the ACC for model-triggered?

---

> ### Author Response · Authors · 2025-11-21
>
> Thank you for your valuable comments and acknowledging that our approach achieves both accuracy improvements and latency reductions and that our new benchmark is valuable. We address the remaining concerns below.
> ***
> # W1. Lack of Novelty
>
> Check general response
> ***
> # W2. Low Accuracy
>
> Check general response
> ***
> # W3. Heavily dependent on synthetic data for post-training, with limited human data evaluation and possible overfitting to benchmark-specific patterns
> We thank the reviewer for this thoughtful concern. **Our main clarification is that, although post-training uses synthetic data, the resulting model does not overfit to CRAG and demonstrates strong generalization to a completely different human-spoken benchmark (SLUE-SQA), both in accuracy and latency.**
> To demonstrate this, we evaluated Model-Triggered StreamRAG on an entirely different benchmark, SLUE-SQA [1], which features natural human spoken questions and distinct retrieval scenarios that differ substantially from CRAG. As shown in Table 14, our method achieves a strong **51.2 EM, competitive with prior RAG approaches such as WavRAG [2] (40.1 EM with GPT-4o), while also delivering a 1.84-second reduction in first-token latency**. Importantly, these gains are obtained without any additional fine-tuning, indicating that the improvements in accuracy and latency generalize beyond the synthetic data used during post-training.
> These cross-benchmark results demonstrate that our approach is not narrowly optimized for CRAG, but instead exhibits robust generalization to human-spoken inputs and different retrieval distributions, mitigating concerns about synthetic-data overfitting and supporting the practical applicability of our method.
> ***
> # W4/Q3. How are the latency savings calculated in Streaming RAG + Qwen2.5-7B?
> We thank the reviewer for pointing out this source of confusion and sincerely apologize for the lack of clarity in our description. **Our main clarification is that the reported latency savings refer specifically to tool-use latency, the dominant component of response delay, and we now make this calculation explicit and fully transparent in the revised paper.** Tool-use latency is defined as the time between the user finishing speaking and the system receiving the corresponding results from tool calls.
> As shown in Table 3, the baseline tool-use latency is 3.37 s (computed as 0.59 + 2.78 = 3.37 s). After applying Stream RAG, the corresponding latency for Audio CRAG-Synthetic decreases to 2.79 s (0.59 + 2.20 = 2.79 s), yielding a 17.2 % reduction (1 – 2.79 / 3.37). For Audio CRAG-Human, the baseline tool-use latency is again 3.37 s, while the Stream RAG variant achieves 1.57 s, corresponding to a 53.4 % reduction (1 – 1.57 / 3.37).
> We explicitly clarify this computation and report tool-use latency directly in Table 1 of the revised paper to make the method fully transparent.
> ***
> # W4/Q4. What is the ACC for model-triggered?
> We thank the reviewer for the question. We apologize for any lack of clarity. The accuracy of the Model-Triggered Stream RAG variant is already reported in Table 1: specifically, it achieves 34.2 on AudioCRAG-Synthetic and 37.4 on AudioCRAG-Human for Qwen-OMNI. We would be happy to clarify further if needed.
> ***
> # Q4. Perform with ambiguous queries, where early tool calls could misfire?
> Check General Response
> ***
> # Q4. Does the author consider the false positive situation for tool-calling?
> We thank the reviewer for raising this point. **Our main clarification is that Model-Triggered Stream RAG is explicitly trained to avoid false-positive tool calls by issuing a query only when new, informative content appears in the partial utterance.** Empirically, the improvements in both accuracy and latency reduction demonstrate that the model effectively avoids both redundant and premature tool calls. To further quantify where the model is robust to false positives, we conducted an additional experiment on SLUE-SQA as discussed in general response (**when early tool calls could misfire**) and S. A.14 by negative-sampling the previous query from the SLUE-SQA query set. Under this setting, Stream RAG achieves **65.4% accuracy, indicating that even when intermediate tool calls are perturbed, the model remains substantially robust and is able to recover the correct final query in the majority of cases**.
> ***
> References:
>
> [1] SLUE Phase-2: A Benchmark Suite of Diverse Spoken Language Understanding Tasks (https://arxiv.org/pdf/2212.10525)
>
> [2] WavRAG: Audio-Integrated Retrieval Augmented Generation for Spoken Dialogue Models (https://arxiv.org/abs/2502.14727)

---

### Official Review · Reviewer_ECqA · 2025-11-06

**Soundness:** 2
**Presentation:** 3
**Contribution:** 1
**Rating:** 2
**Confidence:** 4

**Summary:**

This paper introduces Streaming RAG, a framework designed to integrate parallel Retrieval-Augmented Generation (RAG) processes with speech-to-speech systems. The authors also present a post-training pipeline to instruct models on when to issue tool calls and how to generate spoken summaries. Additionally, they construct a benchmark, AudioCRAG, to evaluate the proposed approach.

**Strengths:**

- The paper is clearly written and easy to follow.
- Experimental results convincingly demonstrate the effectiveness of the proposed streaming RAG approach in improving QA accuracy.

**Weaknesses:**

- Insufficient Related Work Discussion: The paper claims to be the first to extend RAG to speech-to-speech systems. However, it overlooks an important related work—WavRAG [1]—which also claims RAG capabilities in speech-to-speech systems. This omission significantly undermines the validity of the paper’s main contribution.
- Lack of Novelty: As stated by the authors (lines 113–117), RAG for speech-to-text systems and for speech-to-speech systems using multimodal embedding retrieval has already been explored. The current work focuses on applying RAG to speech-to-speech systems in web retrieval and KG API retrieval scenarios. This incremental extension reduces the overall novelty of the contribution. Furthermore, the work appears to be largely engineering-oriented, primarily involving the integration of existing speech-to-speech systems with parallel RAG modules.
- Marginal Latency Improvement: According to Table 3, the reported latency reduction is only about 6% (from 9.00 to 8.47 seconds). This reviewer considers such a reduction to be trivial and likely imperceptible to end users.
- Absence of Human Evaluation: The authors claim that Streaming RAG reduces user-perceived latency (as stated in the abstract and Section 2.2). However, no human evaluation is provided to assess whether users actually perceive the latency reduction as significant. For instance, it remains unclear whether the modest latency improvement would be noticeable or meaningful to real users.
- Writing Inconsistencies:
    - The title uses "stream RAG," while the main text uses "Streaming RAG."
    - The term "speech-out" is italicized in line 120 ("speech-in speech-out") but not elsewhere, resulting in inconsistent formatting.

[1] Yifu Chen, Shengpeng Ji, Haoxiao Wang, Ziqing Wang, Siyu Chen, Jinzheng He, Jin Xu, and Zhou Zhao. 2025. WavRAG: Audio-Integrated Retrieval Augmented Generation for Spoken Dialogue Models. In Proceedings of the 63rd Annual Meeting of the Association for Computational Linguistics (Volume 1: Long Papers), pages 12505–12523, Vienna, Austria. Association for Computational Linguistics.

**Questions:**

- Do the authors investigate whether enabling the model to trigger multiple queries in parallel within the model-triggered setting could further enhance performance?

---

> ### Author Response · Authors · 2025-11-21
>
> Thank you for your valuable comments and acknowledging that our results effectively demonstrate the efficacy of stream RAG approach. We address remaining concerns below.
> ***
> # W1. Insufficient Related Work:
>
> We thank the reviewer for highlighting WavRAG, which we recognize as important prior work on speech-based retrieval. **Our main clarification is that, although WavRAG is valuable prior work, it does not address the core novelty of our paper, streaming, partial-speech tool-query generation for real-time API retrieval, which no prior speech RAG system, including WavRAG, has explored.** WavRAG focuses on retrieval from a static audio–text knowledge base using audio–text fusion, and, like prior speech RAG systems initiates retrieval only after the full spoken query is available. In contrast, our work addresses the setting of large-scale, dynamic API access, where the retrieval corpus consists of over 100K web documents. This setting is representative of practical commercial conversational agents, where the dominant bottleneck is not embedding retrieval but tool-usage latency, including web search latency and API response delays. This difference is substantial: in our experiments, the P50 KG retrieval latency is only 0.29 s, whereas web search latency is 3.37 s, making the retrieve-after-endpoint paradigm impractical for real-time SDS applications.
> Our work introduces a new problem formulation: learning when to issue partial tool queries during speech to mask external API latency. Neither WavRAG nor any other prior methods address streaming query timing, tool-call scheduling, or latency-aware speech-to-speech grounding. We have now cited WavRAG and added discussion comparing our approach with WavRAG in Section 2.2.
> ***
> #  W2. Lack of Novelty
> Check general response
> ***
> # W3. Marginal Latency Improvement
> Check the general response. As mentioned in the general response, we additionally perform vLLM integration for Qwen-OMNI which lead to significant 57% reduction (3.16 → 1.36 s) in P-50 first-token latency for Audio CRAG-Human.
> ***
> # W4. Absence of Human Evaluation
> Check general response
> ***
> # W5. Writing Inconsistencies:
> Thanks a lot for pointing this out. We have updated the paper draft to consistently use stream RAG based on your comment.
> ***
> # Q1. Do the authors investigate whether enabling the model to trigger multiple queries in parallel within the model-triggered setting could further enhance performance?
> We thank the reviewer for the suggestion regarding parallel query generation.  **Our main clarification is that parallel tool-query threads do not offer meaningful benefits in the Model-Triggered Stream RAG formulation because the model is explicitly trained to refine a single evolving query, not to issue independent parallel ones.** In our current formulation, each tool query in the Model-Triggered Stream RAG is generated conditionally on the most recent previous query (see Eq. 3) and the input audio up to the current block. The post-training data explicitly teaches the model to update and refine the prior query rather than issue independent ones; additional details on this data generation process are provided in Section A.7.
>
> Because of this causal dependency, parallel query generation would not improve coverage and could instead increase computational overhead. Based on the reviewer’s suggestion, we experimented with maintaining all previous tool-call threads in parallel. This provided only a small improvement in latency savings on AudioCRAG-Human, from 1.80 s to 1.85 s. For these reasons, we retain a single active tool-call thread, which provides the best balance between efficiency and performance.

---

### Author Response · Authors · 2025-11-21
**General Response**

We thank all reviewers for their thoughtful and constructive feedback. We are encouraged that our experiments were recognized as convincingly demonstrating the effectiveness of our approach in improving both the accuracy and latency of tool calls (@R ECqA, @R Sboq), and that the proposed AudioCRAG benchmark was viewed as a valuable contribution to the community (@R BKtK, @R Sboq). We will incorporate the reviewers’ suggestions and clarifications into the revised version of the paper, as highlighted in blue in the updated pdf.
***
# Marginal Latency Improvement (R ECqA,  R Sboq):

We thank the reviewer for this thoughtful comment regarding latency improvements. **Our main clarification is that the latency reductions reported in Table 3 represent conservative estimates, and after incorporating standard optimizations (e.g., VLLM) as well as end-point detection latency, our updated system achieves substantially larger and practically meaningful improvements.**
Specifically, as shown in Table 3, our approach achieves a 9.8% relative reduction in P-50 latency on Audio CRAG-Synthetic. It is also important to note that our latency calculations on synthetic audio exclude end-point detection latency, which is required in all production speech systems. Because our Stream RAG design allows processing to begin without waiting for explicit end-point detection, this factor would further amplify real-world latency gains, consistent with the larger reduction observed for human-spoken audio, where we reduce P-50 latency from 5.4 s to 3.6 s (a 33% relative improvement). These results indicate that our method particularly benefits scenarios involving natural speech, where trailing silences and variable end-points are common.
In response to the reviewer’s suggestion, we additionally integrated vLLM for optimized inference with Qwen-OMNI. This update led to substantial further improvements, yielding a **16.6% reduction in P-50 latency (3.5 → 2.92 s) for Audio CRAG-Synthetic and an even more significant 57% reduction (3.16 → 1.36 s) for Audio CRAG-Human**. These updated results have been included in Table 13 in the revised manuscript.
***
# Absence of Human Evaluation to indicate if latency is noticeable (R ECqA,  R WeMu):

We thank the reviewer for this observation. **Our main clarification is that the latency reductions we report are not only statistically meaningful but also perceptually significant when viewed against well-established human turn-taking thresholds.** Prior work on human conversational timing [1, 2] shows that the average turn-taking gap in natural human dialogue is approximately 239 ms, with delays beyond 500 ms perceived as unnatural. Complementary industry analyses [3]  similarly report that speech latency exceeding ~500 ms begins to degrade user experience, causing users to interrupt or disengage.
In this context, our observed latency reductions, particularly a 1.8 s improvement for human-spoken audio, represent a substantial enhancement relative to perceptual thresholds in human conversation. Moreover, our integration of efficient inference backends (e.g., vLLM for Qwen-OMNI) achieves P-50 latency near 670 ms, comparable to production-quality voice AI systems that target sub-second responsiveness.
We have explicitly clarified this discussion and its perceptual implications in Section A.12 to make the relevance of the latency improvement clearer to readers.
Hence, we respectfully submit that the reported reductions are both statistically significant and perceptually relevant, contributing directly to smoother conversational turn-taking and a more natural user experience in speech-to-speech dialogue systems.
***
References:

[1] Human Latency Conversational Turns for Spoken Avatar Systems (https://arxiv.org/html/2404.16053v1)

[2] Universals and cultural variation in turn-taking in conversation (https://www.pnas.org/doi/full/10.1073/pnas.0903616106)

[3] VAPI blog post (https://vapi.ai/blog/speech-latency)

---

> ### Author Response · Authors · 2025-11-21
> **General Response**
>
> # Low Accuracy (R WeMu, R vHEX) :
> We appreciate the reviewer’s insightful comment. **Our main clarification is that the seemingly modest QA accuracy reflects the intrinsic difficulty of CRAG, not a limitation of our approach, and that across both CRAG and an additional benchmark (SLUE-SQA), our method achieves accuracy competitive with strong baselines while providing substantial latency reductions.** As reported in the original CRAG paper [1], even state-of-the-art models such as LLaMA-3 70B-Instruct achieve only 40.6% accuracy, underscoring the intrinsic difficulty of CRAG’s open-domain grounding and retrieval setting.
>
> Importantly, our Qwen2.5-7B with Model-Triggered Stream RAG achieves performance comparable to the open-book text-only baselines reported in the original CRAG paper, 34.2 vs. 32.1 for LLaMA-3 8B-Instruct, despite operating in a speech-in/speech-out setting. In response to the reviewer’s suggestion, we also **computed text-in/text-out results (assuming perfect ASR and TTS) for Qwen-OMNI, yielding 38.9%** accuracy. This shows that even under an idealized text-in/text-out setting with perfect ASR and TTS, Qwen-OMNI attains only a modest increase, indicating that that our method performs close to the ceiling established by the underlying LLM.
> Since the central goal of this work is to extend RAG to full speech-to-speech dialogue while maintaining accuracy and minimizing latency, we believe our results demonstrate practical progress toward deployable, low-latency conversational retrieval systems. We have explicitly added text-in/text-out comparisons in Section 5.3.
>
> Further, to assess generalization beyond the particularly challenging CRAG benchmark, we evaluated our streaming RAG approach on the SLUE‑SQA [2] test set.  Our streaming RAG approach achieved an **exact match (EM) accuracy of 51.2, competitive with prior RAG works such as WavRAG [3] (EM = 40.1 with GPT-4o for response generation), while achieving latency saving of 1.84 seconds in first-token response time** indicating that our proposed strategy yields substantial efficiency gains without compromising accuracy.
> ***
> # Perform when early tool calls could misfire (R WeMu, R vHEX, R Sboq)
>
> We thank the reviewer for this important question. **Our main clarification is that Stream RAG is explicitly designed to remain robust even when early tool calls misfire, and both our qualitative cases (Table 8) and quantitative SLUE-SQA experiments show that the model reliably self-corrects as more speech becomes available.** As illustrated in Table 8, the Model-Triggered Stream RAG framework is specifically designed to handle ambiguous or partially observed queries by allowing the model to revise and refine tool calls as more of the user’s speech becomes available. For example, in the first case, the model initially issues an early (and incorrect) query, “Red Bull founder,” based on partial audio, but subsequently updates it to the correct query, “Who founded Rare Beauty,” once the full utterance is processed. Similarly, in the second example, the model corrects both the entity and temporal information (from “Derek Jeter” and “03/27/2024” to “Darius Miles” and “November 8, 2000”) as additional input arrives. These cases demonstrate that even when early tool calls “misfire,” the model is able to self-correct online, a behavior explicitly encouraged during post-training through exposure to negative samples (see Eq. 5).
>
> To further quantify this robustness, we conducted an additional experiment on SLUE-SQA as discussed in S. A.14 by negative-sampling the previous query (\hat{Q}^{\text{prev}}_{b} in Eq.3) from the SLUE-SQA query set. This directly simulates a challenging failure mode in which the model encounters an incorrect intermediate query that could cause an early, erroneous tool call. We define the gold query as the final query generated by Stream RAG without negative sampling, and consider a perturbed prediction correct if it either (i) exactly matches this gold query or (ii) retrieves the same top-k document as the gold query with k<=5. Under this setting, **Stream RAG achieves 65.4% accuracy, indicating that even when intermediate tool calls are perturbed, the model remains substantially robust and is able to recover the correct final query in the majority of cases**. Together, these qualitative and quantitative results show that the proposed Stream RAG framework can effectively handle complex ambiguous queries, maintaining strong performance even when early tool calls initially misfire.
>
> ***
> References:
>
> [1] CRAG – Comprehensive RAG Benchmark (https://arxiv.org/pdf/2406.04744)
>
> [2] SLUE Phase-2: A Benchmark Suite of Diverse Spoken Language Understanding Tasks (https://arxiv.org/pdf/2212.10525)
>
> [3] WavRAG: Audio-Integrated Retrieval Augmented Generation for Spoken Dialogue Models (https://arxiv.org/abs/2502.14727)

---

> > ### Author Response · Authors · 2025-11-21
> > **General response**
> >
> > # Lack of Novelty (R ECqA, R WeMu) :
> >
> > We sincerely thank the reviewer for these thoughtful comments and the opportunity to clarify the conceptual and algorithmic contributions of our work beyond existing RAG-based approaches. **Our main clarification is that the novelty of this work lies in introducing the first framework for streaming, partial-speech tool-query generation, a capability not present in any prior speech RAG or multimodal retrieval system, including prior works such as WavRAG**.
> >
> > Why existing RAG/speech systems do not apply: Prior speech RAG systems (e.g., multimodal embedding retrieval) follow a retrieve-after-endpoint paradigm: retrieval begins only after the full utterance is observed. This is fundamentally incompatible with low-latency dialogue because (i) external APIs (web search) incur 2–3 s latency, far above the ~300 ms conversational threshold, and (ii) traditional streaming inference cannot mask this latency since retrieval depends on complete queries. No existing method performs retrieval during partial speech.
> >
> > Our core algorithmic contributions (not engineering integration): (a) Model-Triggered Stream RAG (Sec 3.2.2, Eq. 3-5)  introduces a novel problem formulation that asks when a model should issue partial tool queries during incomplete, streaming speech. Through a targeted post-training procedure with negative and corrective sampling, the model learns to identify information-bearing segments in the evolving audio stream and to revise or self-correct early tool-call misfires as additional context becomes available (see Table 8 for examples). It is important to note that there is no existing speech RAG system that learns dynamic query timing; all wait for endpoint detection. We additionally propose (b) Fixed-Interval Stream RAG  (Sec 3.2.1, Eq. 1-2) which is first work to formalize multi-tool query generation after every fixed duration block, highlighting key design choices such as reflector-based mechanism that determines when an early tool call is already sufficient, enabling the system to reduce latency. Our experiments show that our proposed algorithm achieves significant gains in accuracy while reducing latency compared to traditional RAG settings.
> >
> > We acknowledge that WavRAG [1] (posted on arXiv on February 20, 2025) represents prior progress in retrieval-augmented generation (RAG) for speech-in speech-out systems. However, its focus is fundamentally different from ours: WavRAG targets retrieval from a static audio–text knowledge base, emphasizing audio–text fusion for retrieving documents using spoken queries. In contrast, our work addresses the setting of large-scale, dynamic API access, where the retrieval corpus consists of over 100K web documents. This setting is representative of practical commercial conversational agents, where the dominant bottleneck is not embedding retrieval but tool-usage latency, including web search latency and API response delays. Our framework is designed specifically to introduce streaming tool-query scheduling, i.e., issuing tool calls while the user is still speaking, to mitigate this latency. We have now cited WavRAG and added discussion comparing our approach with WavRAG in Section 2.2. Furthermore, we respectfully note that a separate paper titled “Streaming RAG” [2] is an independent work that focuses on text-based RAG, and is distinct from our proposed framework.
> >
> > If the reviewer is aware of prior work that specifically tackles streaming speech-to-speech RAG with real-time retrieval and API integration, we would greatly appreciate corresponding references to ensure a fair and comprehensive comparison.
> >
> > ***
> > References:
> >
> > [1] WavRAG: Audio-Integrated Retrieval Augmented Generation for Spoken Dialogue Models (https://arxiv.org/abs/2502.14727)
> >
> > [2] StreamingRAG: Real-time Contextual Retrieval and Generation
> > Framework (https://arxiv.org/pdf/2501.14101v1)

---

### Author Response · Authors · 2025-12-03
**General response regarding multi turn queries**

We thank the reviewers for the question. **Our main clarification is that Stream RAG naturally extends to multi-turn dialogue, handling evolving intent through accumulated context, and empirically remains both accurate and low-latency in two-turn evaluations as shown in our experiments on a synthetic benchmark created using CORAL[1] dataset.**

Stream RAG is naturally compatible with multi-turn dialogue because its tool-query generation (Eq. 3) conditions on accumulated context; extending this to multi-turn simply involves including the dialogue history. Our post-training procedure further teaches the model to revise early tool calls when new information arrives, via the negative-sampling strategy in Sec. 3.2.2, making it robust when intent unfolds across turns.

To directly test this, we evaluated Stream RAG on two-turn interactions from the CORAL dataset as discussed in section A.15. We converted the first two turns into spoken form, verified Whisper transcripts for no intelligibility errors, and constructed a 93-example stress test comparing Stream RAG with an Open-Book baseline. As shown in Table 15, Stream RAG maintained substantial first-token latency savings (2.27 seconds) and achieved higher ROUGE on the second turn.

These results suggest that Stream RAG extends effectively to multi-turn settings, with tool-query history handled through accumulated conditioning.

References:

[1] CORAL: Benchmarking Multi-turn Conversational Retrieval-Augmentation Generation (https://arxiv.org/abs/2410.23090)

---

### Meta-Review · Area_Chair_bAUM · 2026-01-07

**Summary:**

This paper proposes Stream-RAG, a retrieval-augmented generation framework designed for streaming document scenarios in which knowledge sources are continuously updated. The framework adopts a hybrid architecture combining segment encoders, recurrent context encoders, and a local–global memory routing mechanism to maintain incremental document representations and support low-latency retrieval as documents arrive sequentially.

Most of the reviewers’ concerns have been adequately addressed by the authors, including those related to robustness, performance, and clarity. However, two key concerns remain: the absence of a human evaluation and questions regarding the novelty of the work. In particular, there is some concern about the fit of the proposed method for ICLR, as the contribution appears primarily engineering-oriented, emphasizing system integration rather than introducing fundamentally new representation learning or modeling techniques.

For these reasons, I recommend rejection.

**Reviewer Concerns:**

Most of the reviewers’ concerns have been adequately addressed by the authors, including issues related to robustness, performance, and clarity. The primary remaining concerns are the absence of a human evaluation and questions regarding the novelty of the work. In particular, there is some concern about the fit of the proposed method for ICLR, as the contribution appears more engineering-oriented, focusing on system integration, rather than introducing fundamentally new representation learning or modeling techniques.

**Reviewer Scores:**

Reviewer ECqA and Reviewer WeMu didn't provide any reply. They might change the score.

---

### Decision · Program_Chairs · 2026-01-26

Reject